# Decoding Dynamic Visual Experience from Calcium Imaging via Cell-Pattern-Aware Pretraining

**Sangyoon Bae**
Interdisciplinary Program in Artificial Intelligence
Seoul National University
Seoul, 08826, South Korea
{stellasybae}@snu.ac.kr

**Mehdi Azabou**
NSF AI Institute for Artificial and Natural Intelligence (ARNI)
Columbia University
New York City, 10027, United States
{mehdi.azabou}@gmail.com

**Blake Richards**
Mila (Quebec AI Institute)
Dept. of Neurology & Neurosurgery, McGill University
Montreal Neurological Institute, McGill University
School of Computer Science, McGill University
Learning in Machines and Brains Program, CIFAR
Montreal, H2S 3H1, Canada
{blake.richards}@mila.quebec

**Jiook Cha**
Department of Psychology
Department of Brain and Cognitive Sciences
Interdisciplinary Program in Artificial Intelligence
Seoul National University
Seoul, 08826, South Korea
{connectome}@snu.ac.kr

## Abstract

Neural recordings exhibit a distinctive form of heterogeneity rooted in differences in cell types, intrinsic circuit dynamics, and stochastic stimulus–response variability that goes beyond ordinary dataset variability, mixing statistically regular neurons with highly stochastic, stimulus-contingent ones within the same dataset. This heterogeneity poses a challenge for self-supervised learning (SSL)—learnable statistical regularity—thereby destabilizing representation learning and limiting reliable scaling. We introduce POYO-CAP (Cell-pattern Aware Pretraining), a biologically grounded hybrid pretraining strategy that first trains with masked reconstruction plus lightweight auxiliary supervision on statistically regular neurons—identified via skewness and kurtosis—and then finetunes on more stochastic populations. On the Allen Brain Observatory dataset, this curriculum yields 12–13% relative improvements over from-scratch training and enables smooth, monotonic scaling with model size, whereas baselines trained on mixed populations plateau or destabilize. By making statistical predictability an explicit data-selection criterion, POYO-CAP turns neural heterogeneity into a scalable learning advantage for robust neural decoding.

## 1 Introduction

Learning useful representations from neural data poses a fundamental challenge for machine learning, as datasets from varied lab settings are not only small-scale but the signals themselves are complex, highly-dimensional, and only-partially observed (limitation of recording technology), while available labels are typically too scarce and weak for effective supervision. Self-supervised learning (SSL) offers a powerful paradigm to address this data scarcity, as it provides a way to learn from large amounts of data with limited access to labels, thereby allowing many datasets to be combined. This could be particularly useful for reconstructing perceptions or intentions directly from neural activity, e.g. for Brain-Computer Interfaces (BCIs).

However, successful SSL relies on statistical regularities in data, as evidenced by masked modeling and sequence prediction in structured domains such as language (Harris, 1954; Tenney et al., 2019; Sinha et al., 2021; Yu et al., 2024; Lan et al., 2019; Li & Jurafsky, 2017). Neural decoding, in contrast, poses a unique challenge to this prerequisite of predictability. We only record a small,

biased subset of neurons from the full circuit, creating a heterogeneous sample where predictability is not uniform. This unpredictability often correlates with cell type: inhibitory and corticothalamic neurons tend to exhibit more regular dynamics, while excitatory pyramidal cells appear sparser and more stochastic in isolation, partly because we lack access to the broader network signals that drive them. Training SSL models indiscriminately on this mixed-signal data is therefore counterproductive, as the loss becomes dominated by the unpredictable neurons, pulling the model's focus from the relevant and regular patterns it should be learning.

We test the **Statistical Regularity Hypothesis**: that representation learning efficiency scales with the statistical regularity of the selected neural subset. This principle is motivated by the observation that different neural populations, such as inhibitory interneurons and modulatory neurons exhibit fundamentally distinct statistical dynamics. Our hypothesis leads to a "data diet" approach for neuroscience SSL, where, unlike conventional methods that rely on task difficulty, we propose that the intrinsic statistical properties of neurons should guide the learning curriculum.

To validate this, we introduce **POYO-CAP**, a cell-pattern-aware hybrid pretraining framework that uses higher-order statistics (skewness and kurtosis) as proxies for regularity to first pretrain on statistically stable neural populations, combining masked reconstruction with lightweight auxiliary supervision to overcome prior methods' homogeneous treatment of heterogeneous data. Our results confirm the hypothesis: by transforming neural heterogeneity from a challenge into an asset, this approach improves data efficiency by 1.98x and enables high-fidelity movie frame reconstruction from calcium recordings under a controlled stimulus setting, offering a principled, biologically-grounded recipe for scalable neural decoding.

Our contributions are threefold:

- We introduce a biologically-grounded pretraining paradigm that uses statistical regularity (rather than task-based difficulty) to guide data selection, selectively learning from neurons with highly regular responses first before training on more stochastic neurons.
- We present an end-to-end decoder architecture that transforms neural population activity into high-fidelity visual reconstructions, operating independently of external stimulus information.
- We demonstrate that functional heterogeneity, when properly leveraged through our regularity-based data diet, enables robust model scaling unlike conventional approaches that plateau with increased capacity.

**Terminology 1.** We refer to our setup as a *hybrid objective*, a simple form of curriculum learning (Bengio et al. (2009)). The primary objective is masked reconstruction on neural dynamics, while a supervised auxiliary cross-entropy on primitive stimuli serves as an "easy" initial step to stabilize training and prevent representational collapse. Importantly, no downstream labels are used during this pretraining phase.

**Terminology 2.** We define a neuron population as *predictable* from a self-supervised learning (SSL) perspective: its activity must contain sufficient statistical regularity for a model to successfully reconstruct masked portions of its signal. We empirically link this SSL-defined predictability to low skewness and kurtosis in calcium traces. Thus, while our definition aligns with the neuroscientific concept of stable firing patterns, it remains a fundamentally operational one, tied to the success of the masked reconstruction task.

## 2 RELATED WORK

**Decoding Models for Neuroscience**   Recent neural decoding models span diverse architectures and learning paradigms. Transformer-based approaches such as POYO (Azabou et al., 2023) and POYO+ (Azabou et al., 2024) enable multi-session learning but depend on full supervision, limiting scalability to unlabeled data. Self-supervised methods like CEBRA (Schneider et al., 2023) relax label requirements for single-session training but require labels for multi-session training. In visual reconstruction, fMRI-based frameworks have reached high fidelity citepchen2023cinematic, joo2024brain through masked modeling and large generative models, but rely on indirect stimulus-to-brain mappings from fMRI's slow hemodynamic signal. While these approaches set benchmarks for fMRI, direct comparison is challenging due to modality differences. In contrast, our method learns directly from neural recordings using intrinsic population dynamics with minimal auxiliary supervision.

**SSL in Neuroscience**   Most self-supervised approaches to neural data assume population homogeneity and ignore functional specialization. Models such as Neuro-BERT Wu et al. (2022) treat all neurons equally, while contrastive or task-aware methods Song et al. (2023); Zhao et al. (2024) depend on external supervision rather than intrinsic circuit structure. These frameworks overlook that predictable neurons (inhibitory interneurons and modulatory pathways) differ fundamentally from stimulus-encoding neurons in computational role and temporal dynamics. Recent work by Johnson et al. (2022) characterized such heterogeneity through in vivo imaging, while our predictability-based selection offers distinct computational advantages by identifying and pretraining on regulatory neurons, enabling SSL to capture circuit-level dynamics and improving scalability beyond uniform population models.

**Data-Centric SSL and Neural Heterogeneity**   Our approach aligns with the emerging "data diet" perspective in machine learning, which posits that the quality of pre-training data is as critical as its quantity (Paul et al., 2021; Zhuang et al., 2025). However, we distinguish our framework from these methods in a fundamental way: while standard approaches prune training *samples* (e.g., specific images or text), our strategy selects *neurons* (feature sources). In neural recordings, heterogeneity is intrinsic to the sensor array itself, not just the examples. We demonstrate that adding more neurons can paradoxically lead to a "scaling collapse"—a failure mode unique to heterogeneous neural populations. By selecting neurons based on statistical regularity, we resolve this collapse and transform heterogeneity from a liability into an asset for scaling.

## 3   METHODS

### 3.1   DATASET AND PARTITION

We use the Allen Brain Observatory (BO) calcium imaging dataset, featuring recordings from 13 Cre driver lines (de Vries et al., 2020). We partition the dataset into pretraining and finetuning sets at the Cre-line level. To form the pretraining set, we identified a "predictable" subset by applying a knee-detection algorithm (Algorithm S1) to the per-line skewness and kurtosis distributions. This a priori process selected four lines (SST, VIP, PVALB, and NTSR1) that fell below the statistical knee—corresponding to major inhibitory interneuron classes and one modulatory excitatory line. The remaining Cre lines were reserved for finetuning and downstream evaluation. For downstream tasks (movie reconstruction and drifting grating decoding), train/validation/test splits were performed at the trial level within each session. Specifically, stimulus trials from each recording session (i.e., each animal) were divided into non-overlapping temporal segments for training, validation, and testing. Thus, the same animal may appear across splits, but distinct trials were used for evaluation, preventing temporal leakage while assessing generalization across neural responses to unseen stimulus segments. Importantly, pretraining and finetuning datasets were strictly separated at the animal level due to the disjoint Cre-line partitioning: animals used for pretraining (predictable lines) do not overlap with those used for finetuning (unpredictable lines). This design ensures cross-subject transfer at the pretraining stage, while downstream evaluation focuses on generalization across trials within each biological subject. To guarantee a fair comparison, we verified that all models (including baselines and ablations) were evaluated on identical held-out trial splits. Details are described in Appendix A.

### 3.2   CELL-PATTERN-AWARE PRETRAINING

#### 3.2.1   DATA-EFFICIENT SELECTION CRITERIA

We hypothesize that neurons showing **statistical regularity** are ideal for effective SSL pretraining. Within our framework, we operationally define this as *predictability*—the inherent structure enabling effective masked reconstruction. To identify these neurons without labels, we leverage per-neuron **skewness** and **kurtosis**. We refer to the selected subset as exhibiting **near-Gaussian activity** (mean skewness 1.87, kurtosis 7.32), characterized by symmetric, **thin-tailed** distributions suitable for learning general features. In stark contrast, excluded neurons exhibit **heavy-tailed**, sparse bursting (mean kurtosis 148.51), better reserved for task-specific fine-tuning. For rigorous empirical validation of these metrics, see Appendix C.

| **Pretraining** | |
| --- | --- |
| Data Size | 134 sessions, 80,146 samples |
| Selection Criteria | $skewness \leq 3.51$, $kurtosis \leq 22.62$ |
| Hardware | 4×V100 (KISTI `cas_v100nv_4`) |
| **Fine-tuning (Movie decoding, Drifting Gratings)** | |
| Selected Data | 299 sessions, 1,170,931 samples |
| Selection Criteria | $skewness > 3.51$, $kurtosis > 22.62$ |
| Frames (movie decoding) | 900 |
| Hardware | 4×V100 (KISTI `cas_v100nv_4`) |

Table 1: **Computationally-Efficient Pretraining** Summary of dataset scale (sessions and samples), predictable-neuron selection criteria (skewness and kurtosis computed on per-neuron $\Delta F/F$ traces over the full recording), and computational setup for pretraining and fine-tuning.
**Notes.** (1) Selected Data = number of predictable (pretraining) / unpredictable (finetuning) sessions / samples after skewness/kurtosis filtering. (2) For movie decoding, training batches preserved temporal order, whereas validation and test batches were randomly shuffled to evaluate generalization beyond temporal continuity.

To objectively partition the data, we applied a **knee-detection algorithm** (Satopaa et al. (2011)) to find a data-driven threshold across the 13 discrete CRE lines. Specifically, we identified the knee point on the sorted distribution of per-line mean statistics, establishing a cutoff based on cell-type categories rather than individual neuron scores. While this approach failed for lower-order statistics like event rate and Fano factor, it revealed a clear breakpoint for both skewness and kurtosis, providing a principled basis for our split. The resulting data-driven thresholds (skewness $\leq 3.51$, kurtosis $\leq 22.62$) identified a "predictable" subset comprising four CRE lines: **SST, VIP, PVALB, and NTSR1**. This statistically derived group is also biologically coherent, consisting of three major inhibitory interneuron classes and one regulatory corticothalamic excitatory line (NTSR1), all of which are crucial for stabilizing neural circuits. This convergence of statistical and biological criteria validates that our method effectively captures neurons showing statistically regular firing pattern. Crucially, these thresholds were determined *a priori* as a single, fixed criterion to partition the dataset, not as a tunable hyperparameter, which is why a sensitivity analysis was not performed.

### 3.2.2 Model Framework

**Predictable Neuron Pretraining with Auxiliary Classification** We introduce a latent masked modeling approach to train our model: masked and an unmasked views of the same sample are fed independently through the encoder, the latent representation of the unmasked view is then used as target for the latent representation of the masked variant. To avoid representational collapse (Grill et al. (2020); Chen et al. (2020)), we use a supervised auxiliary loss. This auxiliary loss *bootstraps* early selectivity while masking-based reconstruction *shapes* representations for downstream decoding. The primitive labels also serve as guidance to stabilize early optimization.

Our architecture is based on the POYO+ (Azabou et al. (2024)) architecture: calcium traces are tokenized into a sequence of input tokens that are then compressed, using a cross-attention block, into a sequence of latent tokens, which we note $Z_1 = \{\mathbf{z_1^{(1)}}, \cdots, \mathbf{z_1^{(L)}}\}$, where $L$ is the number of latent tokens and $\mathbf{z_1^{(i)}} \in \mathbb{R}^d$ is the latent embedding. Each latent token $\mathbf{z_1^{(i)}}$ has an associated timestamp relative to the context window. We introduce the following temporal masking scheme: we causally mask a percentage of the latent tokens to form $Z_1^{\mathrm{masked}} = \{\mathbf{z_1^{(1)}}, \cdots, \mathbf{z_1^{(L-M)}}, <\mathrm{MASKED}> \cdots, <\mathrm{MASKED}>\}$. We selected a masking ratio of 50% empirically, i.e. the second half of the context window is masked. We use a siamese network (see Figure 1) to feed both $Z_1$ and $Z_1^{\mathrm{masked}}$ through the same self-attention blocks which yields $Z_L$ and $Z_L^{\mathrm{masked}}$ respectively. Finally, we use $Z_L$ as the target for $Z_L^{\mathrm{masked}}$.

During pre-training, the model is trained on a joint objective, consisting of self-supervised masked reconstruction and fully-supervised classification of drifting grating orientations. This auxiliary classification task stabilizes the early training dynamics before the model focuses on the complex downstream movie decoding task.

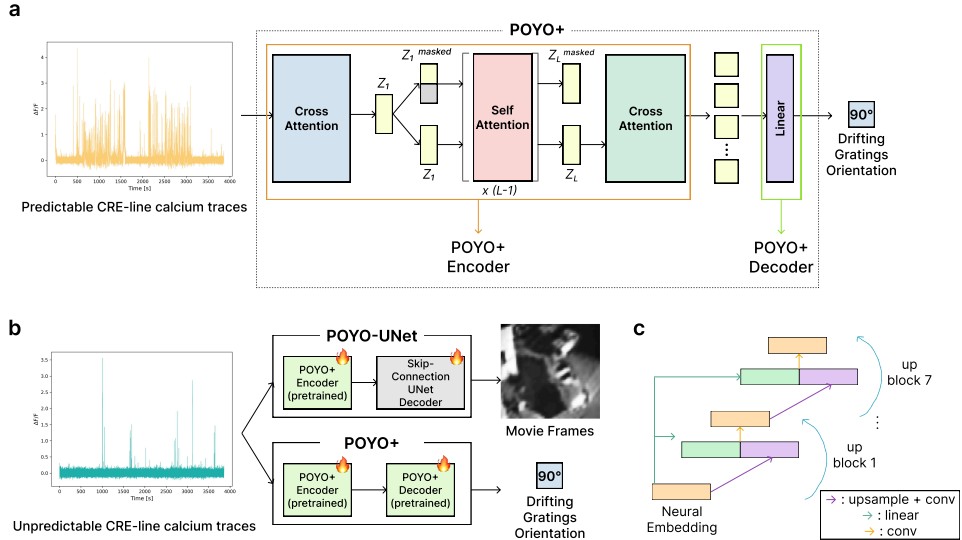

Figure 1: **Overall Framework of POYO-CAP. (a)** Pretraining strategy using predictable calcium traces with masked reconstruction learning (50% masking on temporal dimension). **(b)** Task-specific finetuning with unpredictable traces using either skip-connection UNet decoder (complex tasks) or original POYO+ decoder (simple tasks). **(c)** Skip-Connection UNet Decoder architecture replacing traditional encoder skip connections with neural embedding projections.

The pre-training loss is as follows:

$$\text{Loss}_{\text{pretrain}} = \text{Loss}_{\text{L1}}(Z_L^{\text{masked}}, Z_L) + \lambda \cdot \text{Loss}_{\text{CrossEntropy}}(\text{DG}_{\text{predicted}}, \text{DG}_{\text{true}}) \quad (1)$$

where $\lambda$ is a loss weight that we empirically found $\lambda = 0.01$ to be optimal through grid search ($\lambda \in 0.001, 0.01, 0.1$, with performance degrading by 7-11% for $\lambda < 0.001$ or $\lambda > 0.1$. We keep the cross-entropy weight small so CE accelerates convergence while masking drives representation formation. This hybrid objective operationalizes a curriculum learning strategy, where the simple auxiliary task provides a stable foundation for the more demanding masked reconstruction objective. Details are provided in Appendix E.

**Task-Specific Fine-tuning on Unpredictable Neurons** Finetuning uses unpredictable CRE-line traces with task-specific decoders. For classification and simple regression tasks such as drifting-grating orientation prediction, we use the POYO+ multi-task decoder, and for complex movie frame reconstruction we employ a dedicated vision-specialized Skip-Connection U-Net decoder.

The finetuning loss is as follows:

$$Loss_{movie} = 50\,Loss_{focal} + 50\,Loss_{L1} + 50\,Loss_{FFT} + Loss_{perceptual} + 0.1\,Loss_{SSIM} \quad (2)$$

$$Loss_{DG} = Loss_{CrossEntropy}(DG_{\text{predicted}}, DG_{\text{true}}) \quad (3)$$

Loss weights in Eq. 2 were determined through a systematic grid search over [0.1-100] using SSIM validation score. The different loss terms in the movie reconstruction loss corresponds to specialized components (Focal (Lin et al. (2017)), FFT (Fast Fourier Transform, (Zhao et al. (2016))), Perceptual (Johnson et al. (2016)), and SSIM (Wang et al. (2004))) that ensure high-fidelity image reconstruction. See Appendix I for details on each loss term.

**Skip-Connection U-Net Decoder** To address the challenge of reconstructing high-resolution movie frames, we designed a specializaed decoder, as this dense prediction task requires custom vision modules that were not designed in the POYO+ decoder. Our new U-Net-inspired decoder generates frames from a single neural embedding. In each upsampling stage, a direct projection of the latent vector (e.g., to $128 \times 2 \times 2$, $64 \times 4 \times 4$) is concatenated with the upsampled feature map and fused with a $1 \times 1$ convolution. These repeated latent injections are crucial for maintaining semantic information across all scales, enabling the faithful reconstruction of fine visual details from a compact neural representation. See Appendix H for more details.

### 3.3 NUMERICAL ANALYSIS

#### 3.3.1 LOSS LANDSCAPE ANALYSIS

To understand the challenge of optimizing representation learning models on neural data, we projected neural activity onto its first two principal components (PCs) and approximated the reconstruction loss landscape. The loss at each grid point in the PC space was estimated using a k-nearest neighbor approach ($k = 5$), which considered the local variance of nearby data points and a distance penalty term. Landscapes were smoothed for visualization via a Gaussian filter ($\sigma = 1.0$) (Li et al. (2018)).

#### 3.3.2 INFORMATION THEORY ANALYSIS

We used Fisher Information as a metric for data quality, where $I(\theta) = \mathbb{E}\left[\left(\frac{\partial}{\partial\theta}\log p(x|\theta)\right)^2\right]$ quantifies the amount of information each data point provides about underlying model parameters Amari (1998), with higher values indicating better parameter estimation and convergence. For a quasi-Gaussian signal, this can be approximated as the inverse of the signal variance ($I \approx 1/\sigma^2$). Based on this, we defined the **Effective Dataset Size** ($D_{eff}$) as the raw data size weighted by its quality, where a higher Fisher Information value corresponds to a larger effective size. This allows for a more accurate comparison of dataset utility beyond simple data point counts (Kaplan et al. (2020)).

### 3.4 REPRESENTATION ANALYSIS

We quantified the properties of the learned latent spaces using several metrics, including t-SNE for visualization, Intrinsic Dimension (ID) for efficiency (Levina & Bickel (2004)), and metrics to assess geometric dissimilarity and structural integrity. To assess dissimilarity between latent spaces learned by different models, we used Procrustes disparity (Dryden & Mardia (2016)) and Centered Kernel Alignment (CKA) (Kornblith et al. (2019)). To evaluate local structure, we used a Temporal Neighborhood Preservation score (Venna & Kaski (2001)) (see Appendix J for all definitions).

## 4 RESULTS

### 4.1 EXPERIMENTAL SETUPS AND BASELINES

To isolate the benefits of our cell-pattern-aware pre-training, we compare our main model, **POYO-CAP**, against a crucial baseline:

- **Supervised Baseline (From-Scratch):** To rigorously quantify the performance gains from our SSL stage, we compare against a baseline sharing an identical encoder–decoder architecture but trained end-to-end on the downstream tasks without pre-training.

- **Architecture Ablation Studies:** To disentangle the contributions of encoder representations and decoder capacity, we include three capacity-matched variants. Capacity matched means total parameters are within $\pm3\%$ of our model.: (i) *MLP Encoder → MLP Decoder*, which maps neurons directly to pixels through a deep fully-connected network with no spatial inductive bias; (ii) *POYO Encoder → MLP Decoder*, which retains our SSL encoder but replaces the U-Net decoder with a purely linear decoder to test whether learned representations alone can drive performance; (iii) *POYO Encoder → U-Net Decoder without skip connections*, which preserves the U-Net hierarchy but removes lateral skip pathways to assess the importance of multiscale feature fusion.

We compare to POYO+ (Azabou et al., 2024) which is a state-of-the-art model. To benchmark against external SSL methods, we evaluated an adapted CEBRA baseline (Schneider et al., 2023) by training its encoder and feeding representations to our vision decoder. This yielded an SSIM of $\sim$0.48, confirming that contrastive latent spaces optimized for behavioral alignment do not transfer effectively to high-fidelity pixel generation. For CEBRA, we report the best performance between training from scratch and fine-tuning strategies. Regarding Neuro-BERT (Wu et al., 2022), the lack of an official implementation prevented a reproducible adaptation, and thus it was excluded.

## 4.2 EFFECT OF CELL-PATTERN-AWARE PRETRAINING

### 4.2.1 CELL-PATTERN-AWARE PRETRAINING ENABLES SMOOTH LOSS LANDSCAPE

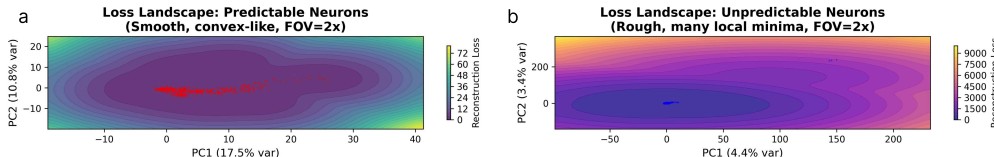

Figure 2: **Loss Landscape Topology Reveals a Dichotomy in Optimization Difficulty.** Masked reconstruction loss landscapes for predictable and unpredictable neurons, projected onto their first two principal components (PCs) with an expanded field of view (FOV=2x). **(a)** The landscape from predictable neurons is smooth and convex-like, clearly revealing high-loss boundaries that enclose data points (red dots) in a single basin, indicating a well-posed optimization problem. **(b)** In contrast, the landscape from unpredictable neurons is rugged and non-convex, characterized by numerous local minima, which presents a challenging, ill-posed problem.

Our analysis of the masked reconstruction loss landscape elucidates a fundamental dichotomy in the nature of the optimization problems presented by the two neural populations. Predictable neurons induce a geometrically well-posed landscape characterized by a smooth, convex-like surface (roughness $\sigma_L = 14.8546$), which is highly amenable to gradient-based optimization methods. In stark contrast, unpredictable neurons give rise to a treacherous, non-convex landscape (roughness $\sigma_L = 2048.4712$) plagued by a multitude of spurious local minima. Crucially, the quantitative contrast remains striking even with the expanded FOV: despite the inclusion of steep basin walls, the 'unpredictable' landscape remains $\sim 138\times$ rougher than the 'predictable' one. This confirms that our conclusion is robust to the choice of scale: the structural optimization gap between the two populations is massive, regardless of the field of view. This topological difference explains why the pre-training task transforms from a simple optimization challenge to a complex, ill-posed problem, thereby providing a rigorous geometric basis for the superior performance of the predictable-first pre-training curriculum.

### 4.2.2 PREDICTABLE NEURONS CONTAINS RICHER REPRESENTATION

| Metric | Predictable | Unpredictable | Ratio (Pred./Unpred.) |
|---|---|---|---|
| Fisher Information (Data Quality) | 64.51±0.55/-0.65 | 33.47±0.46/-0.35 | **1.93x** |
| Data Quality Ratio (Efficiency) | 34.41 | 17.39 | **1.98x** |
| Effective Dataset Size | 71.5 M | 227.5 M | - |

Table 2: **Information-Theoretic Analysis of Data Quality.** A quantitative comparison of predictable and unpredictable neural populations. The analysis reveals that predictable data is information-theoretically superior, providing a basis for its enhanced performance and scalability. Values are reported as mean ± 95% CI.

Our analysis revealed that predictable neural data is information-theoretically richer, which translates directly to greater data efficiency. We quantified this using **Fisher Information**, finding that the predictable dataset had a value of 64.5 compared to 33.5 for the unpredictable dataset, indicating that each predictable data point contains **1.93 times more information** for model training (Table 2). Consequently, while the raw dataset sizes were comparable, the quality-adjusted **Effective Dataset Size** ($D_{eff}$) was significantly larger for the predictable population, making each of its data points 1.98 times more efficient for training.

### 4.2.3 CELL-PATTERN-AWARE PRETRAINING ACHIEVES HIGH PERFORMANCE

As shown in Table 3, our cell-pattern-aware pretraining yields consistent performance gains across downstream tasks. On the movie decoding task, our approach achieves an SSIM of 0.593 for neural-to-visual reconstruction (Figure 3). We note that training and test frames were drawn from the same stimulus set; thus, the task evaluates generalization of neural-to-frame mappings across temporal segments and pretraining transfer, rather than reconstruction of entirely novel stimuli. On the

| Method | Pretrain Data | Finetune Data | Movie SSIM↑ | DG Accuracy↑ |
|---|---|---|---|---|
| **POYO-CAP (Ours)** | Predictable | Unpredictable | **0.593±0.013** | **0.555±0.022** |
| **Baseline: Train on All** | N/A (From Scratch) | All (Pred. + Unpred.) | 0.528±0.023 | 0.492±0.041 |
| *Architecture Ablation Studies* | | | | |
| MLP Enc.→MLP Dec. | Predictable | Unpredictable | 0.449±0.022 | – |
| POYO+ Enc.→MLP Dec. | Predictable | Unpredictable | 0.503±0.019 | – |
| POYO+ Enc.→UNet Dec. without skip connection | Predictable | Unpredictable | 0.466±0.047 | – |
| CEBRA Enc.→UNet Dec. | Predictable | Unpredictable | 0.481±0.010 | – |
| *Data-Selection Ablation Studies* | | | | |
| Inhibitory-only SSL | Inhibitory | Excitatory | 0.544±0.030 | 0.537±0.025 |
| Reverse SSL | Unpredictable | Predictable | 0.489±0.032 | 0.213±0.037 |
| Mixed SSL | Unpred. + partial Pred. | Unpredictable | 0.543±0.049 | 0.313±0.012 |
| Random subset SSL | Random (Size-matched) | Remaining | 0.532±0.044 | 0.254±0.011 |
| *Pretraining Objective Ablation Studies* | | | | |
| Random Masking Loss | Predictable | Unpredictable | 0.540±0.017 | 0.548±0.028 |
| Masking Loss only | Predictable | Unpredictable | 0.496±0.050 | 0.099±0.019 |
| Large CE weight (0.1) | Predictable | Unpredictable | 0.552±0.052 | 0.482±0.033 |
| Small CE weight (0.001) | Predictable | Unpredictable | 0.532±0.042 | 0.469±0.015 |
| Cross-Entropy Loss only | Predictable | Unpredictable | 0.506±0.057 | 0.452±0.026 |

Table 3: **Performance comparison across multiple visual decoding tasks.** Our proposed framework, POYO-CAP, consistently outperforms baseline models, demonstrating the effectiveness and generalizability of cell-pattern-aware self-supervised learning. Best results are shown in bold. Movie decoding task is denoted as movie, drifting gratings decoding task is denoted as DG. Values are depicted as mean $\pm$ 95% CI across three seeds (with $p<0.05$ (paired t-test)). Dashes indicate tasks not applicable to image-only decoders.

drifting-gratings classification task, our model reaches 55.5% accuracy, outperforming the from-scratch baseline (49.2%).

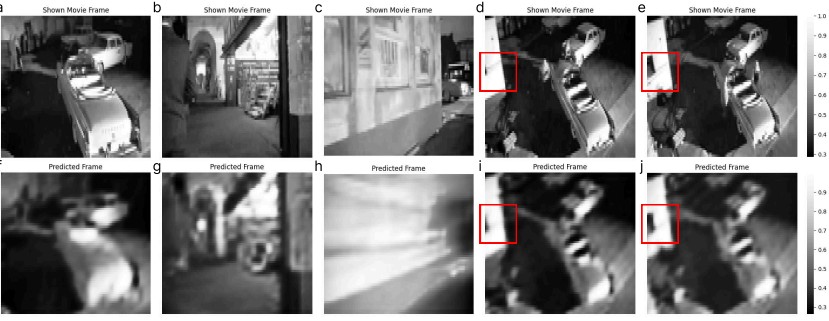

Figure 3: **End-to-end neural-to-vision decoding**. **(a-e)** depict movie frames presented to the mouse, **(f-j)** depict reconstructed frames. Our model captures subtle frame-to-frame variations (red boxes), demonstrating faithful reconstruction of stimulus-driven visual structure rather than trivial averaging.

Ablation studies highlight the benefits of our approach, indicating that both the architecture and the learning objective tailored to the data's statistics are important factors. The superior performance of temporal masking over random masking underscores the value of the objective and lends functional support to our selection criteria. Temporal masking preserves local temporal dependencies critical for neural dynamics (typically 50-100ms receptive fields in V1 neurons), while random masking disrupts these patterns. This result suggests the curated neurons ("predictable" neurons) indeed possess the predictable temporal structure that a specialized task can effectively exploit. Furthermore, data-selection ablations indicate that data quality can outweigh quantity; reversing the curriculum to pretrain on unpredictable neurons leads to worse performance than training from scratch, sug-

gesting that pretraining on highly stochastic data may establish a less effective inductive bias for downstream learning. Overall, our approach of selectively pretraining on neurons with regular firing patterns leverages population heterogeneity to enable stable and scalable representation learning.

### 4.2.4 THE REPRESENTATIONAL ADVANTAGE OF CELL-PATTERN-AWARE PRETRAINING

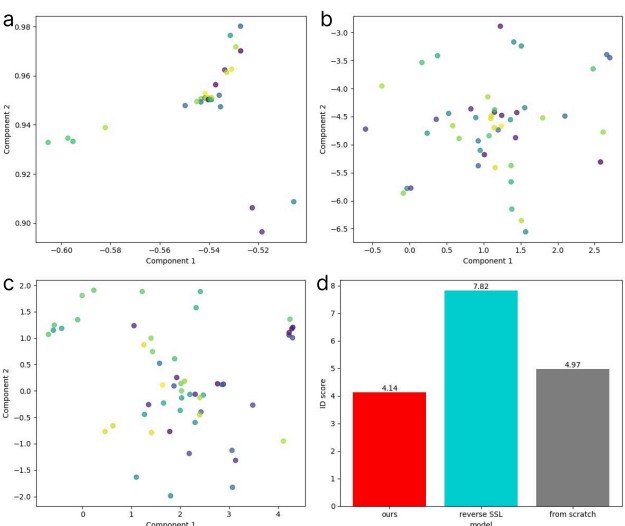

Figure 4: **POYO-CAP learns a more efficient and structured latent manifold. (a-c)** t-SNE visualization of latent spaces from our POYO-CAP model (a), a reverse SSL model (b), and a from-scratch model (c). Point color reflects the temporal progression of frames. **(d)** Quantitative comparison of the Intrinsic Dimension (ID) for each model.

Analysis of the learned representations reveals a stark contrast between the strategies (Figure 4). Qualitatively, t-SNE visualizations show that our POYO-CAP model learns a well-structured manifold that captures the data's temporal continuity, while baseline approaches like reverse SSL and from-scratch training yield disorganized or collapsed representations. This visual observation is validated by multiple quantitative metrics. Our model's latent space is more efficient, with a significantly lower intrinsic dimension (ID) of 4.14 compared to the from-scratch (4.97) and reverse SSL (7.82) models. It also better preserves local temporal structure, evidenced by a higher Temporal Neighborhood Preservation score (0.2355 vs. 0.1584 and 0.0960). Furthermore, high Procrustes disparity ($>0.98$) and low Centered Kernel Alignment (CKA,$\approx0.13$) confirm that the methods learn fundamentally different feature spaces. Taken together, these results demonstrate that our selective pre-training is crucial for learning a concise and structured representation of the neural code.

### 4.3 POYO-CAP ENABLES STABLE MODEL SCALING

A key advantage of our framework is its ability to enable stable model scaling, a critical property for building more powerful decoders. To quantify this, we performed a bootstrap regression analysis ($N = 10,000$). While models trained from scratch (gray) and those pretrained on only unpredictable neurons (cyan) exhibit erratic or flat scaling (slopes $\approx$ 0.005–0.013), our main approach (red) unlocks consistent performance gains as model capacity increases, achieving a statistically significant positive slope (0.018, $p < 0.01$). This represents a $\sim$40% steeper scaling trajectory compared to the from-scratch baseline. This demonstrates that a well-designed pre-training strategy is a prerequisite for effective scaling.

Furthermore, comparing pre-training data mixtures reveals what constitutes a good pre-training set. The model pretrained on mixed predictable and unpredictable neurons (yellow) excels at smaller scales but fails to improve at larger capacities (slope=0.006). This suggests that the quality, not merely the quantity, of pre-training data is the key determinant for scalability. In our view, the noisy signals from unpredictable neurons function as an information bottleneck, limiting the formation of a robust and smoothly scalable representation. Conversely, pre-training on the "clean" signal

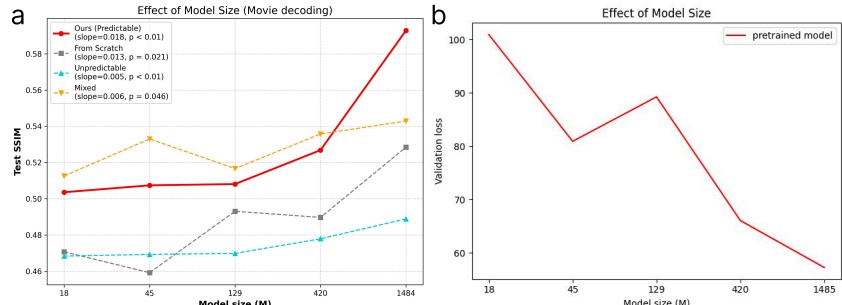

Figure 5: **Pre-training with predictable neurons is crucial for effective model scaling.** **(a)** Test SSIM performance versus model size for different pre-training strategies. Only the model pretrained exclusively on predictable neurons (red) demonstrates robust, positive scaling with model capacity (slope=0.018, $p < 0.01$ under bootstrap analysis). In contrast, training from scratch (gray) or including unpredictable neurons in pre-training (cyan, yellow) leads to flat or erratic scaling (slopes $\approx$ 0.005–0.013). **(b)** Corresponding validation loss during pre-training on the predictable set, showing a general downward trend that indicates successful learning.

from predictable neurons (red) builds a superior foundation that larger models can exploit, leading to significant performance gains. This successful scaling is corroborated by the general decrease in validation loss during the pre-training stage, as shown in Figure 5b.

## 4.4 MECHANISTIC ANALYSIS OF TRANSFER

To understand the transfer from predictable to unpredictable neurons, we analyzed parameter dynamics during fine-tuning. Pretraining establishes a stable "representational scaffold" that is largely preserved: encoder weights change by only $\sim$0.18% (norm $\approx$ 222,909), whereas the readout layer adapts substantially, with bias magnitudes increasing 12.4$\times$ ($p < 0.01$). This suggests that the encoder provides a pre-optimized latent manifold (Figure 2), while the readout layer adjusts task-specific decision boundaries without destabilizing the representation (see Appendix F for details).

## 5 CONCLUSION

We introduce POYO-CAP, a cell-pattern-aware hybrid pretraining framework that makes self-supervised learning (SSL) robust to the brain's statistical complexity. Neural recordings combine multiple dynamical regimes—statistically regular activity and heavy-tailed bursts—and indiscriminate SSL on this mixture can ill-condition optimization and limit reliable scaling.

POYO-CAP treats statistical predictability as an explicit data-selection principle. Using skewness and kurtosis, we identify a predictable subset and pretrain with temporal masked reconstruction plus lightweight auxiliary supervision, then fine-tune on more irregular populations. On the Allen Brain Observatory calcium imaging dataset, this predictable-first curriculum improves decoding (movie reconstruction SSIM = 0.593; drifting gratings accuracy = 0.555), increases effective data efficiency by 1.98$\times$, and yields smooth, monotonic gains with model capacity where baselines plateau or destabilize. The skewness/kurtosis thresholds serve as computational proxies consistent with known biological distinctions without implying a causal mechanism, and our evaluation focuses on mouse visual cortex under controlled stimulus conditions. These results frame predictability-first as a testable curriculum signal for scaling neural SSL beyond this setting.

ACKNOWLEDGEMENTS

This work was supported by the National Research Foundation of Korea (NRF) grant funded by the Korea government (MSIT) (No. 2021R1C1C1006503, RS-2023-00266787, RS-2023-00265406, RS-2024-00421268, RS-2024-00342301, RS-2024-00435727, RS-2025-25457239, RS-2021-NR061370, NRF-2021M3E5D2A01022515, and NRF-2021S1A3A2A02090597), by the Researchers Program through Seoul National University (No. 200-20250071, 200-20250049, 200-20250116, 200-20250115, 200-20250113). Additional support was provided by the Institute of Information & Communications Technology Planning & Evaluation (IITP) grant funded by the Korea government (MSIT) [No. RS-2021-II211343, Artificial Intelligence Graduate School Program, Seoul National University] and by the Global Research Support Program in the Digital Field (RS-2024-00421268). This work was also supported by the Artificial Intelligence Industrial Convergence Cluster Development Project funded by the Ministry of Science and ICT and Gwangju Metropolitan City, by the Korea Brain Research Institute (KBRI) basic research program (25-BR-05-01), by the Korea Health Industry Development Institute (KHIDI) and the Ministry of Health and Welfare, Republic of Korea (HR22C1605), and by the Korea Basic Science Institute (National Research Facilities and Equipment Center) grant funded by the Ministry of Education (RS-2024-00435727). We acknowledge the National Supercomputing Center for providing supercomputing resources and technical support (KSC-2023-CRE-0568, KSC-2024-CRE-0198, KSC-2025-CRE-0340).

An award for computer time was provided by the U.S. Department of Energy's (DOE) ASCR Leadership Computing Challenge (ALCC). This research used resources of the National Energy Research Scientific Computing Center (NERSC), a DOE Office of Science User Facility, under ALCC award m4750-2024, and supporting resources at the Argonne and Oak Ridge Leadership Computing Facilities, U.S. DOE Office of Science user facilities at Argonne National Laboratory and Oak Ridge National Laboratory.

This work was supported by the funds provided by the National Science Foundation and by DoD OUSD (R&E) under Cooperative Agreement DBI-2229929 (The NSF AI Institute for Artificial and Natural Intelligence).

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

## A  DETAILED DATA SPLITTING AND EMBEDDING INITIALIZATION

### A.1  TRIAL-LEVEL TRAIN/VALIDATION/TEST SPLITS

For downstream tasks (movie reconstruction and drifting gratings decoding), train/validation/test splits were performed **within each session** (i.e., within each animal recording) at the trial level.

Specifically, stimulus presentations were temporally divided into non-overlapping trial segments independently for each session:

- **Natural Movie One:** 80% train / 10% validation / 10% test (shuffle = False; temporal order preserved)
- **Drifting Gratings:** 70% train / 10% validation / 20% test (shuffle = True)

Thus, the same animal may appear in train/validation/test splits, but distinct, non-overlapping stimulus trials are used. This design prevents temporal leakage while evaluating generalization across unseen stimulus segments within each biological subject. Importantly, downstream evaluation does *not* involve animal-level train/test separation.

### A.2  PRETRAINING–FINETUNING CROSS-ANIMAL SEPARATION

In contrast to downstream splits, pretraining and finetuning datasets are strictly separated at the animal level.

In the Allen Brain Observatory dataset, each animal belongs to a single Cre driver line. Pretraining uses sessions from inhibitory/modulatory Cre lines (SST, VIP, PVALB, NTSR1), whereas finetuning uses sessions from excitatory Cre lines. Because Cre lines are disjoint at the animal level, the pretraining and finetuning animal sets are completely non-overlapping.

This Cre-line-based partition corresponds to the "held-out animals" referenced in our reviewer responses: animals used during finetuning were never seen during pretraining.

### A.3  UNIT EMBEDDING INITIALIZATION AND TRANSFER

Because pretraining and finetuning use disjoint animal sets, their recorded neurons do not overlap. Consequently, pretrained unit embeddings cannot be reused during finetuning.

During finetuning, all unit embeddings are randomly initialized from $\mathcal{N}(0, 0.02)$.

Only the encoder (Perceiver) weights are transferred from pretraining. During finetuning, **all parameters**—including the newly initialized unit embeddings and the transferred encoder weights—are trained end-to-end.

Thus, cross-animal transfer arises solely from shared encoder representations rather than reused neuron-specific embeddings.

## B  DESIGN CHOICES AND BASELINE SELECTION

We focus our evaluation on architectures with comparable capacity for high-dimensional visual reconstruction. Many recent SSL methods in neuroscience are designed for different objectives. For instance, while contrastive methods like CEBRA (Schneider et al. (2023)) are effective for behavioral alignment, our empirical evaluation confirmed that their low-dimensional embeddings are suboptimal for direct pixel-level generation. Similarly, masked autoencoding methods such as Neuro-BERT (Wu et al. (2022)) were excluded due to the lack of an official implementation and insufficient architectural capacity for high-resolution image generation. We therefore selected POYO+ (Azabou et al. (2024)) as our primary comparative model for its flexible architecture that can be scaled for dense prediction tasks.

To support our visual reconstruction objective ($304 \times 608$ pixel images), we scaled the architecture to use 1024-dimensional embeddings, a substantial increase from the 64 dimensions used in the original work for classification. This architectural parity ensures a fair comparison: both our method

and the from-scratch baseline operate with identical encoder-decoder capacity. This design choice allows us to isolate the contribution of our cell-pattern-aware SSL approach from architectural advantages, providing a rigorous evaluation of our core hypothesis.

## C  JUSTIFICATION FOR DATA PARTITIONING CRITERIA

### C.1  DETAILED DESCRIPTION ON CRE LINES

| Cre Line | Type | Functional Role |
|---|---|---|
| EMX1 | Excitatory | Pan-excitatory, broad cortical excitatory neurons |
| SLC17A7 | Excitatory | Pan-excitatory, glutamatergic projection neurons |
| CUX2 | Excitatory | Upper layer excitatory, intracortical connections |
| RORB | Excitatory | Layer 4 excitatory, thalamic input recipients |
| SCNN1A | Excitatory | Layer 4 excitatory, primary sensory processing |
| NR5A1 | Excitatory | Layer 4 excitatory, sensory feature detection |
| RBP4 | Excitatory | Layer 5 excitatory, subcortical projections |
| FEZF2 | Excitatory | Deep layer excitatory, long-range projections |
| TLX3 | Excitatory | Layer 5 excitatory, corticotectal projections |
| NTSR1 | Excitatory | Layer 6 excitatory, corticothalamic feedback |
| VIP | Inhibitory | Disinhibitory interneurons, modulate inhibition |
| SST | Inhibitory | Somatostatin interneurons, lateral inhibition |
| PVALB | Inhibitory | Parvalbumin interneurons, fast spiking, timing |

Table S1: Cre driver lines in the Allen Brain Observatory dataset

This table provides detailed information on the 13 Cre driver lines from the Allen Brain Observatory dataset used in this study. A central premise of our work is that the heterogeneous nature of neural populations is a critical factor for self-supervised learning. This table offers a comprehensive overview of this heterogeneity by detailing the specific functional roles and types of the neuronal subpopulations available in the dataset.

Each Cre Line targets a specific type of neuron based on the expression of a particular gene, allowing for cell-type-specific measurements. These are broadly categorized into two main **Types**:

- **Excitatory neurons**: These neurons, such as `Emx1` and `Slc17a7`, typically release neurotransmitters like glutamate that increase the likelihood of a postsynaptic neuron firing. As detailed in the **Functional Role** column, they are involved in a wide range of activities, from broad cortical activation to specific roles in sensory processing (e.g., `Scnn1a` in Layer 4) and forming long-range projections to other brain areas (e.g., `Rbp4` in Layer 5).

- **Inhibitory neurons**: These interneurons, such as `SST` and `Pvalb`, typically release neurotransmitters like GABA that decrease the likelihood of a postsynaptic neuron firing. Their functional roles are often modulatory, involved in processes like lateral inhibition (`SST`), network disinhibition (`Vip`), and regulating the precise timing of neural activity (`Pvalb`).

As described in the main text, our data-driven selection method identified four lines (`SST`, `VIP`, `PVALB`, and `NTSR1`) from this diverse catalog as having the 'predictable' dynamics suitable for our pre-training objectives. This table provides the full context for that selection, detailing the characteristics of all potential cell types considered in this work.

### C.2  VALIDATION OF SKEWNESS AND KURTOSIS AS PREDICTABILITY INDICATORS

**Justification for Higher-Order Statistics**  Our central hypothesis is that neurons with different functional roles exhibit distinct statistical signatures in their activity patterns. To create a principled data partition for our curriculum, we sought metrics that could reliably separate these populations. This analysis provides the empirical justification for our choice of skewness and kurtosis over simpler, lower-order statistics.

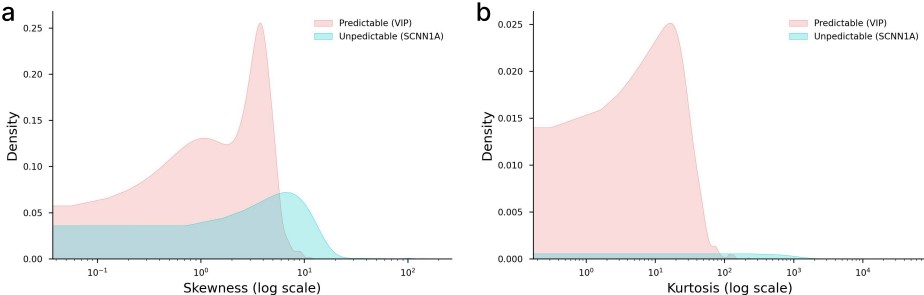

Figure S1: Statistical distributions of predictable and unpredictable neural subpopulations. Kernel Density Estimate (KDE) plots for **(a)** skewness and **(b)** kurtosis of calcium traces. The "Predictable" group (pink), selected for our pre-training, exhibits distributions sharply concentrated at low values for both metrics. In contrast, the "Unpredictable" group (cyan) shows broad, heavy-tailed distributions. This clear statistical separation validates our data-driven criteria for identifying stable neuron populations suitable for self-supervised learning. This statistical separation captures sub-types of neurons that may have more regulatory functions: inhibitory interneurons (SST, VIP, PVALB) and modulatory excitatory neurons (NTSR1), which all may be more involved in network stabilization rather than stimulus-specific responses.

**Interpreting the Metrics in a Neuroscience Context**   In the context of calcium imaging traces, skewness and kurtosis serve as powerful proxies for the temporal dynamics of a neuron's activity:

- **Skewness** measures the asymmetry of the activity distribution. A low skewness (close to zero) implies a symmetric, quasi-Gaussian distribution, characteristic of neurons with stable baseline activity that fluctuates evenly. In contrast, a high positive skewness indicates a distribution with a long right tail, the statistical fingerprint of a neuron that is mostly quiescent but fires in sparse, high-amplitude positive bursts.

- **Kurtosis** measures the "tailedness" of the distribution, or the prevalence of extreme outliers. Low kurtosis is characteristic of Gaussian-like activity. High kurtosis indicates a "spiky" or leptokurtic distribution, where extreme events (large calcium transients) are far more common than would be expected from random noise. This is a hallmark of event-driven, stimulus-encoding neurons.

**Empirical Validation of Statistical Separation**   The distributions shown in Figure S1 confirm that these metrics provide a clear and robust separation between our two target populations.

- **Panel (a)** shows that the 'Predictable' group (pink) has a skewness distribution sharply peaked at low values, consistent with symmetric activity patterns. The 'Unpredictable' group (cyan), however, is broadly distributed across much higher skewness values, confirming a burst-like firing pattern.

- **Panel (b)** reveals an even starker separation for kurtosis. The 'Predictable' group's distribution is almost entirely concentrated at low values, indicating a near-total absence of extreme outlier events. This provides strong evidence that these neurons exhibit highly regular and constrained dynamics.

**Functional Interpretation**   This clear statistical separation aligns directly with the known functional roles of the underlying neuron types. The low-skew, low-kurtosis profile is the statistical signature of neurons engaged in network stabilization and modulation—the very neurons we identify as 'predictable' (SST, VIP, PVALB, NTSR1). Conversely, the high-skew, high-kurtosis profile is the classic signature of sparse, stimulus-encoding neurons that fire selectively and powerfully. This strong correspondence between a data-driven statistical signature and a known biological function validates our selection criteria as a principled method for identifying ideal neuron candidates for self-supervised pre-training.

**How predictable lines were chosen.** For each of the 13 CRE lines, skewness and kurtosis were computed from its neural activity distribution before training. A single knee (NTSR1) was estimated on the per-line statistic distribution, yielding four predictable lines used entirely for pretraining; the remaining lines were reserved exclusively for finetuning/validation/test. This is a line-level dataset split; no animals/sessions/neurons overlap across partitions.

| CRE Line | Number of Cells | Event Rate | | Fano Value | |
|---|---|---|---|---|---|
| | | Median | Std | Median | Std |
| EXM1 IRES CRE | 7537 | 1.021 | 0.122 | 103869.897 | 1611.701 |
| SLC17A7 IRES2 CRE | 7736 | 1.046 | 0.149 | 103676.897 | 2260.452 |
| CUX2 CREERT2 | 10275 | 1.034 | 0.182 | 103686.898 | 2314.886 |
| RORB IRES2 CRE | 5009 | 1.055 | 0.291 | 103464.896 | 3461.491 |
| SCNN1A TG3 CRE | 1200 | 1.078 | 0.221 | 103217.894 | 2426.047 |
| NR5A1 CRE | 2135 | 1.125 | 0.361 | 102710.887 | 4346.653 |
| RBP4 CRE KL100 | 1611 | 1.121 | 0.237 | 102770.890 | 2962.409 |
| FEZF2 CREER | 587 | 1.079 | 0.142 | 103497.896 | 2182.647 |
| TLX3 CRE PL56 | 1524 | 1.075 | 0.126 | 103190.893 | 1473.551 |
| NTSR1 CRE GN220 | 1239 | 1.041 | 0.0981 | 103566.895 | 1149.701 |
| VIP IRES CRE | 639 | 1.379 | 0.309 | 99914.863 | 3951.127 |
| SST IRES CRE | 573 | 1.183 | 0.240 | 101967.881 | 2844.855 |
| PVALB IRES CRE | 245 | 1.332 | 0.308 | 100983.849 | 3995.032 |

Table S2: Event rate and Fano value statistics for each CRE line

In this section, we provide the empirical justification for selecting skewness and kurtosis as the primary statistical indicators for identifying predictable neural subpopulations. We conducted a comparative statistical analysis of the calcium trace signals between the predictable and unpredictable neuron groups, as defined by the criteria in the main text. The results are summarized in Table S2.

As shown in Table S2, first and second-order statistics, namely the mean and variance of the activity, showed no statistically significant differences between the two populations ($p=0.347$ and $p=0.281$, respectively). This suggests that simpler metrics related to the overall magnitude or spread of neural activity are insufficient to distinguish between neurons with different response pattern regularities.

| CRE Line | Number of Cells | Skewness | | Kurtosis | |
|---|---|---|---|---|---|
| | | Median | Std | Median | Std |
| EXM1 IRES CRE | 7537 | 5.637 | 6.169 | 88.966 | 887.759 |
| SLC17A7 IRES2 CRE | 7736 | 5.132 | 4.380 | 63.847 | 132.297 |
| CUX2 CREERT2 | 10275 | 5.504 | 4.644 | 79.245 | 186.898 |
| RORB IRES2 CRE | 5009 | 6.283 | 5.300 | 88.748 | 443.990 |
| SCNN1A TG3 CRE | 1200 | 7.240 | 15.235 | 103.458 | 3027.682 |
| NR5A1 CRE | 2135 | 6.159 | 8.254 | 69.922 | 1286.154 |
| RBP4 CRE KL100 | 1611 | 7.395 | 14.528 | 94.758 | 2377.191 |
| FEZF2 CREER | 587 | 5.108 | 3.763 | 55.862 | 96.430 |
| TLX3 CRE PL56 | 1524 | 6.133 | 3.910 | 76.118 | 105.617 |
| NTSR1 CRE GN220 | 1239 | 2.453 | 3.579 | 22.616 | 83.209 |
| VIP IRES CRE | 639 | 3.507 | 1.770 | 19.145 | 22.122 |
| SST IRES CRE | 573 | 2.075 | 3.007 | 12.932 | 259.785 |
| PVALB IRES CRE | 245 | 1.991 | 1.525 | 8.258 | 17.978 |

Table S3: Skewness and kurtosis statistics for each CRE line

In stark contrast, higher-order statistics (Table S3) revealed dramatic and highly significant differences. The predictable subpopulation exhibited low average skewness (1.87) and kurtosis (7.32), characteristic of more symmetric and less outlier-prone signal distributions. Conversely, the unpredictable subpopulation showed extremely high average skewness (9.84) and kurtosis (148.51),

indicating heavily right-tailed and sparse, spiky activity patterns. These differences were statistically significant to a very high degree ($p < 0.001$).

This analysis empirically confirms that skewness and kurtosis are exceptionally effective and reliable indicators for differentiating neural populations based on their activity patterns, far more so than lower-order statistics. This provides a strong validation for our methodological choice to use these metrics as the core selection criteria within the POYO-CAP framework.

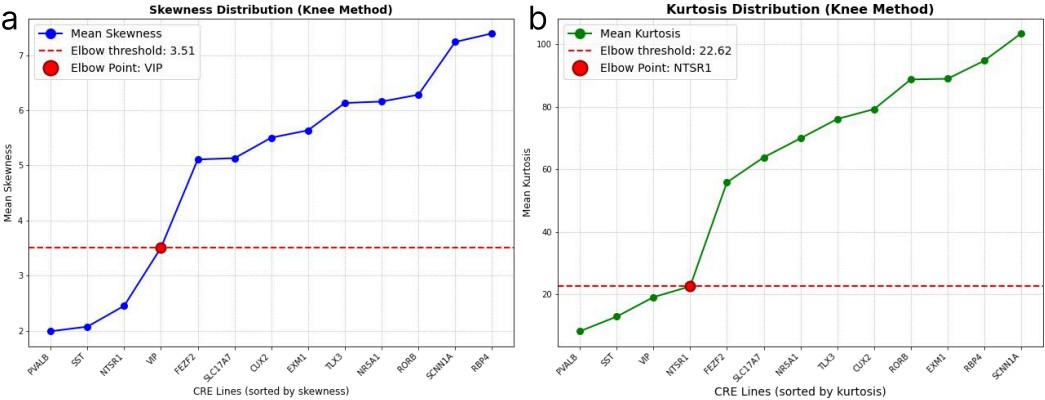

Figure S2: **Data-driven threshold determination for predictable neuron selection.** **(a)** Distribution of mean skewness values across CRE lines, sorted in ascending order. The knee detection algorithm identified a natural breakpoint at skewness = 3.51 (red dashed line), corresponding to the NTSR1 CRE line (red circle). CRE lines below this threshold exhibit stable, near-Gaussian activity patterns suitable for self-supervised pretraining. **(b)** Distribution of mean kurtosis values across CRE lines, showing a similar elbow at kurtosis = 22.62 (red dashed line), again at the NTSR1 boundary. The sharp increases beyond these breakpoints indicate the transition from predictable regulatory neurons to highly variable, stimulus-contingent populations. This objective approach ensures biologically grounded selection criteria rather than arbitrary thresholds.

---

**Algorithm S1** Find Elbow (Knee) Point by Maximum Gradient

---

**Input:** Vector of values $y = [y_1, y_2, \ldots, y_n]$
**Output:** Knee index $k$
    Compute consecutive gradients $g_i \leftarrow y_{i+1} - y_i$ for $i = 1, \ldots, n-1$
    Find index $k \leftarrow \arg\max_i g_i$   ▷ position of largest gradient
    **return** $k$   ▷ knee is the point **before** the sharpest rise

---

To objectively determine the threshold values for predictable neuron selection, we employed a knee detection algorithm on the distribution of skewness and kurtosis values across CRE lines. For each metric, we calculated the gradient between consecutive CRE lines (sorted by their respective mean values) and identified the point preceding the sharpest increase as the elbow point (See algorithm S1). This approach revealed natural breakpoints at skewness $\leq 3.51$ and kurtosis $\leq 22.62$, corresponding to the NTSR1 CRE line as the boundary case (Figure S2). CRE lines below these thresholds (SST, VIP, PVALB, and NTSR1) exhibited consistently low and stable activity statistics, while those above showed sharp increases indicative of more variable, stimulus-driven responses. This data-driven approach ensures that our selection criteria are grounded in the natural distribution of neural activity patterns rather than arbitrary cutoffs, providing an objective foundation for distinguishing predictable from unpredictable neural subpopulations.

Note: CRE line labels were only used to define the domain-level split (which lines go to pretraining vs. finetuning) and were not used inside training losses, model selection, or evaluation.

# D    THEORETICAL JUSTIFICATION FOR PRIORITIZING PREDICTABLE NEURONS IN PRE-TRAINING

To understand the mechanisms behind the improved performance of our SSL methodology, we conducted a theoretical and empirical analysis comparing the properties of two representative neural populations: 'Predictable' (VIP inhibitory neurons) and 'Unpredictable' (Scnn1a excitatory neurons). This analysis reveals that the statistical and temporal characteristics of 'Predictable' neurons create a more favorable learning scenario for SSL models.

## D.1    ENHANCED TEMPORAL STRUCTURE AND INFORMATION CONTENT

Self-supervised learning on time-series data fundamentally relies on exploiting temporal regularities. Our analysis shows that 'Predictable' neurons possess a much richer and more stable temporal structure.

**Temporal Predictability**    As shown in the autocorrelation plot (**Fig. S3d**), the signal from predictable neurons maintains a stronger correlation with its recent past compared to unpredictable neurons. This slower decay indicates that each time point contains more information about its neighbors, providing a more robust signal for temporal contrastive learning tasks (Oord et al. (2018)).

**Reconstruction Fidelity**    From an information theory perspective, signals that are easier to compress and reconstruct are more amenable to representation learning. We quantified this using the Cramér-Rao Lower Bound (CRLB), a theoretical minimum for estimator variance (Kay (1993)). The analysis (**Fig. S3c**) shows that the mean CRLB for predictable neurons is 0.0476, while it is 0.1443 for unpredictable neurons. This suggests that **predictable neurons can be reconstructed with 3.03 times greater theoretical efficiency**, providing a more reliable learning signal with lower intrinsic noise.

**Signal Dynamics**    The power spectrum (**Fig. S3e**) reveals that predictable signals have their power concentrated in low-frequency bands, indicative of smooth and continuous dynamics (Buzsaki & Draguhn (2004)). In contrast, unpredictable signals have a flatter power spectrum, closer to white noise, signifying less discernible temporal structure.

## D.2    FAVORABLE STATISTICAL DISTRIBUTIONS AND LEARNING STABILITY

Beyond temporal structure, the underlying statistical distribution of the data dramatically impacts the stability and efficiency of the learning process, particularly for gradient-based optimization.

**Distributional Properties**    The activity of unpredictable neurons follows a sparse, heavy-tailed distribution, as visualized in the histogram (**Fig. S3b**). This is quantitatively confirmed in **Fig. S3h**, where these neurons exhibit extreme skewness (mean: 10.65) and kurtosis (mean: 475.93). Such distributions, with rare but high-amplitude events, can lead to unstable gradients and cause the model to be overly influenced by outliers (Gurbuzbalaban et al. (2021)). In contrast, the predictable neurons are quasi-Gaussian (mean skewness: 2.56, mean kurtosis: 12.98), providing a more well-behaved statistical foundation for learning.

**Learning Stability**    We analyzed the stability of the learning problem by computing the condition number of the data's covariance matrix, which reflects the curvature of the loss landscape. A high condition number implies a landscape with sharp, narrow valleys, making it difficult for optimizers to converge (Nocedal & Wright (2006)). The condition number for unpredictable neurons was 627.49, whereas it was only 67.15 for predictable neurons (**Fig. S3i**). This demonstrates that the learning problem posed by **unpredictable neurons is approximately 9.34 times more ill-conditioned, or harder to optimize**, than that of predictable neurons.

## D.3    SYNTHESIS: PREDICTED SSL EFFICIENCY

**Methodology for Composite Score**    To synthesize these multifaceted properties into a single metric, we formulated a composite score for predicted SSL efficiency. This score is a weighted average

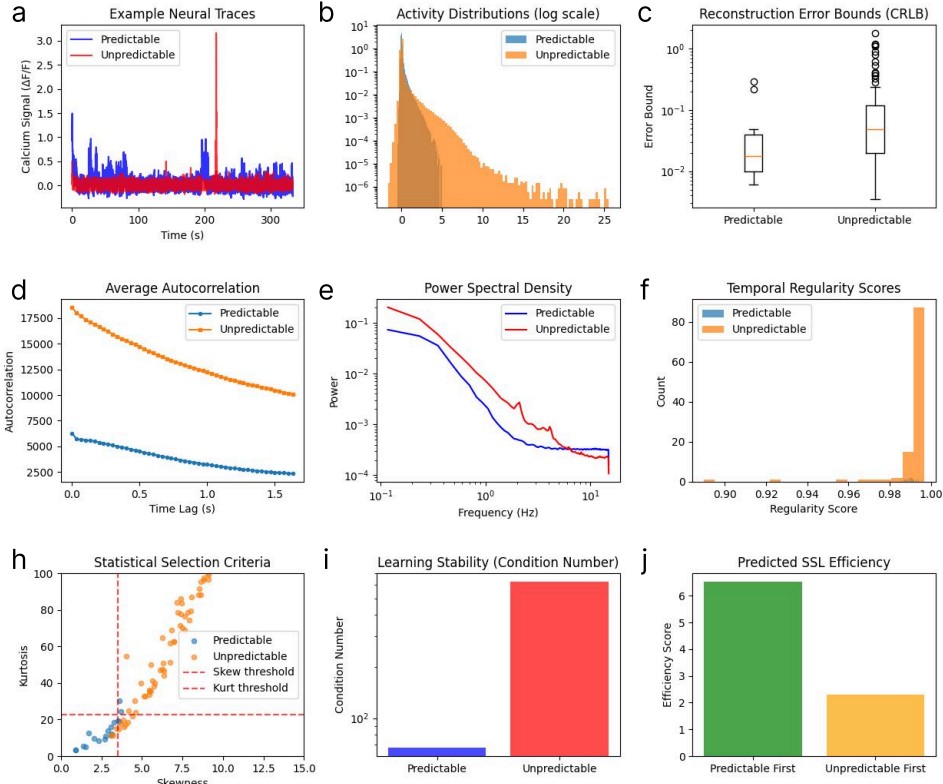

Figure S3: **Theoretical Analysis of Neural Signal Properties for Self-Supervised Learning (SSL) Efficiency.** This figure provides a comprehensive comparison between two distinct types of neural activity: 'Predictable' signals derived from inhibitory VIP neurons, which exhibit quasi-Gaussian distributions, and 'Unpredictable' signals from excitatory Scnn1a neurons, characterized by sparse, skewed distributions. The analysis dissects why 'Predictable' neurons serve as a more effective dataset for SSL pre-training. **(a)** Example calcium signal traces ($\Delta F/F$) over 350 seconds. The predictable trace (blue) shows smoother fluctuations, while the unpredictable trace (red) is characterized by sparse, high-amplitude bursts. **(b)** Log-scale histograms of signal activity distributions, highlighting the heavy-tailed, skewed nature of unpredictable signals compared to the more centered predictable signals. **(c)** Boxplot of the theoretical reconstruction error bounds (Cramér-Rao Lower Bound, CRLB). Predictable neurons show a significantly lower and tighter error distribution, indicating they are more reliably encoded. **(d)** Average autocorrelation functions. Predictable signals exhibit a slower decay in autocorrelation, signifying more persistent temporal structure. **(e)** Power spectral density (PSD) analysis. Predictable signals have more power concentrated at lower frequencies, consistent with smoother dynamics. **(f)** Distribution of temporal regularity scores. **(h)** Scatter plot of kurtosis versus skewness for individual neurons. Predictable neurons (blue) largely fall within the statistical selection criteria (red dashed lines), whereas unpredictable neurons (orange) do not. **(i)** Learning stability, quantified by the condition number of the data covariance matrix. The much higher condition number for unpredictable data indicates a more ill-conditioned and unstable learning problem. **(j)** A composite score predicting overall SSL pre-training efficiency, integrating metrics from the preceding panels. Pre-training with predictable data first is predicted to be substantially more efficient.

of five key factors derived from our preceding analyses: (1) **Reconstruction Fidelity**, based on the inverse of the theoretical error bound (CRLB); (2) **Learning Stability**, derived from the inverse of the learning problem's condition number; (3) **Temporal Regularity**, measured by the signal's autocorrelation and consistency; (4) **Information Content**, based on signal entropy; and (5) **Favorable Statistical Properties**, rewarding low skewness and kurtosis. These factors were weighted (0.3, 0.25, 0.2, 0.15, and 0.1, respectively) to reflect their relative importance in creating a learnable, information-rich dataset.

**Predicted Efficiency and Rationale** The resulting composite score (**Fig. S3j**) predicts that pre-training on a dataset of predictable neurons first is **2.85 times more efficient** than starting with unpredictable neurons. This theoretical result strongly supports our empirical findings and provides a clear rationale for our pre-training strategy: by first learning from the stable, information-rich, and well-conditioned 'predictable' neurons, the model can establish a robust foundational representation before being fine-tuned on more complex, sparse signals.

## E    THEORETICAL JUSTIFICATION FOR CURRICULUM LEARNING

To provide a theoretical basis for our hybrid pre-training objective, particularly the use of a simple auxiliary task (drifting gratings) as a warm-up, we conducted a simulation of curriculum learning principles. We defined sample difficulty based on local variance, distance to the manifold center, and local density, and simulated four training strategies: Easy-to-Hard, Hard-to-Easy, Random, and Mixed.

The results, summarized in Figure S4, unequivocally support an Easy-to-Hard curriculum. This strategy led to the highest final performance for both data types, achieving **1.43x better results for predictable data and 1.19x for unpredictable data** compared to a random ordering, while also ensuring superior training stability. Notably, the absolute performance and stability achieved on predictable data (0.9967 performance, 0.9998 stability) were substantially higher than on the more volatile unpredictable data (0.6945 performance, 0.9340 stability). This highlights that while an optimal curriculum is always beneficial, the intrinsic quality of the "easy" examples ultimately determines the robustness of the learned foundation. This analysis provides a principled foundation for our training methodology, where the simple DG task serves as the initial "easy" stage that stabilizes the model, and the broader predictable-first pre-training represents a macro-level application of the same principle.

## F    DETAILED WEIGHT DYNAMICS ANALYSIS

To empirically validate the "representational scaffold" hypothesis, we analyzed the model parameters before and after fine-tuning. We computed the relative $L_2$ norm of the weight changes in the PerceiverIO encoder versus the task-specific readout heads.

**Encoder Stability.** The encoder, responsible for mapping neural activity to the latent space, showed minimal change during the fine-tuning phase on unpredictable neurons. The relative weight change was 0.183%, indicating that the features learned from the predictable subset are robust and generalizable to the broader population. The stability of encoder norms ($\approx$222,909) suggests the model stays within the same optimization basin found during pre-training ((Garipov et al., 2018)).

**Readout Adaptation.** Conversely, the readout layer demonstrated dramatic specialization. The magnitude of the readout biases increased $12.4\times$. This confirms that transfer occurs through optimization geometry ((Neyshabur et al., 2020)): the encoder maintains the stable manifold, while the readout adapts to the specific statistics and noise profile of the unpredictable neurons.

## G    A UNIFIED THEORETICAL FRAMEWORK FOR POYO-CAP

Our empirical results, particularly the successful scaling of our model, are underpinned by a cohesive theoretical framework derived from the principles of representation and curriculum learning. This framework explains why the strategic use of neural heterogeneity is not merely an effective heuristic but a principled approach to building scalable models of neural dynamics.

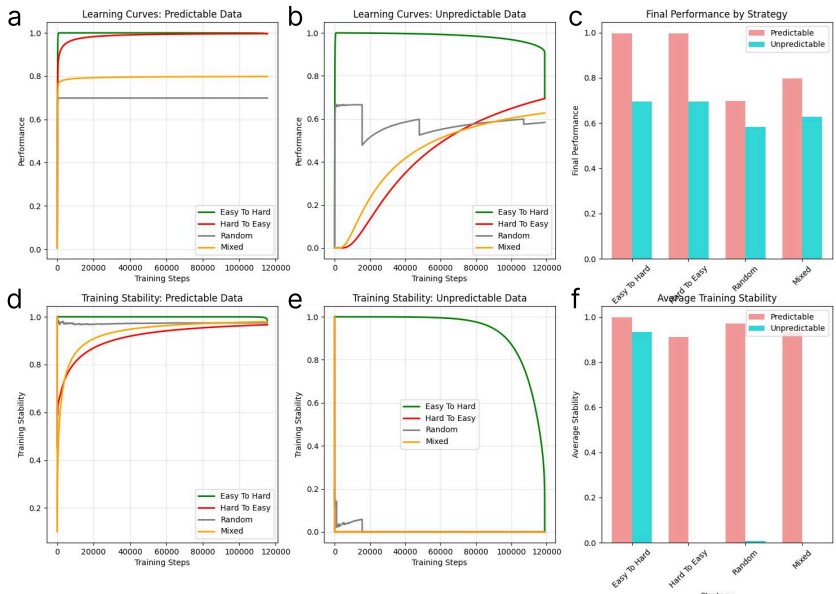

Figure S4: **Theoretical Simulation of Curriculum Learning Strategies.** This figure simulates the effect of different data ordering strategies on model performance and training stability for both predictable and unpredictable neural data. **(a, b, c)** Learning curves and final performance comparison. The "Easy to Hard" curriculum (green) achieves the fastest convergence and highest final performance. **(d, e, f)** Training stability analysis. The "Easy to Hard" strategy maintains high stability, while the "Hard to Easy" approach (red) suffers from significant initial instability. These results provide a strong theoretical justification for our predictable-first, curriculum-based pre-training strategy.

**The Representational Advantage of Predictable Data** The success of any learning algorithm is contingent on the quality of the data representation. Our analysis reveals that predictable neurons provide a fundamentally superior substrate for representation learning. They induce a smooth, convex-like loss landscape (Fig. 2), which makes optimization a well-posed problem. Furthermore, the representations learned from this data are more efficient and structured, evidenced by their significantly lower intrinsic dimension and more organized latent manifold (Fig. S3). This efficiency is rooted in their higher information content, as quantified by Fisher Information (Table 2), allowing the model to learn a robust representation from a smaller effective dataset size.

**The Optimization Advantage of a Predictable-First Curriculum** Beyond the static quality of the data, the order of presentation is critical. Our theoretical simulations of curriculum learning (Fig. S4) demonstrate that an "Easy-to-Hard" strategy is optimal, maximizing both final performance and training stability. Starting with easy examples—those with clear, low-variance signals—allows the model to establish a stable foundational representation. The predictable neurons, with their inherent statistical regularity, serve as the ideal "easy" examples in the context of neural data.

**Synthesis: The Synergy of Representation and Curriculum** The remarkable success and scalability of POYO-CAP can be understood as a direct result of the synergy between these two principles. Our method does not merely use a better curriculum; it applies the **optimal curriculum to the optimal data**. By starting with predictable neurons, we solve a well-posed representation learning problem in a maximally stable manner. This establishes a robust initial model that is well-prepared to subsequently learn the fine-grained, complex features from the unpredictable data during fine-tuning. This unified view provides a rigorous mathematical and conceptual foundation for our empirical scaling results (Fig. 5), explaining why POYO-CAP unlocks consistent performance gains with increasing model capacity while other approaches stagnate or fail.

---

**Algorithm S2** UNet Decoder with Latent Injection

---

**Input:** Latent vector $z \in \mathbb{R}^d$
**Output:** Reconstructed frame $\hat{x} \in \mathbb{R}^{64 \times 128}$

$x \leftarrow \text{reshape}(z, [d, 1, 1])$
**for** each upsampling stage $i = 1, \ldots, 4$ **do**
$\quad s_i \leftarrow \text{Linear}(z) \rightarrow \text{reshape}([c_i, h_i, w_i])$
$\quad x \leftarrow \text{Upsample}(x, \text{scale} = 2)$
$\quad x \leftarrow \text{Conv2d}(x)$
$\quad x \leftarrow \text{concat}([x, s_i])$
$\quad x \leftarrow \text{Conv2d}_{1 \times 1}(x)$
$\hat{x} \leftarrow \text{ExtraUp}(x) \quad \triangleright 32^2 \rightarrow 64 \times 128$

---

## H  SKIP-CONNECTION UNET DECODER ARCHITECTURE

## I  SPECIALIZED LOSS COMPONENTS

To ensure high-fidelity image reconstruction, we employ a composite loss function with several specialized components. We adapt Focal Loss to a regression task to emphasize challenging pixels and refine fine details (Eq. 4). $\alpha$ and $\gamma$ are set as 1 empirically. To preserve high-frequency structure, we introduce a frequency-domain loss using the Fast Fourier Transform (Eq. 5). Perceptual similarity is further promoted through both an SSIM loss (Eq. 6) and a perceptual loss computed as the mean–squared error (MSE) between feature maps of an ImageNet-pretrained AlexNet.

Specifically, we extract activations from the first four convolutional blocks of the AlexNet (Krizhevsky et al. (2012)) feature extractor ('layer=3' in the PyTorch implementation) after ImageNet normalization (mean $[0.485, 0.456, 0.406]$, standard deviation $[0.229, 0.224, 0.225]$) (Eq. 7).

## J  REPRESENTATION ANALYSIS

For qualitative visualization, high-dimensional latent embeddings were projected into a two-dimensional space using t-SNE (t-distributed Stochastic Neighbor Embedding). We then quantified the global properties of these spaces using three metrics: (1) Intrinsic Dimension (ID) to measure the efficiency of the representation, (2) Procrustes disparity, and (3) Centered Kernel Alignment (CKA) to assess the geometric dissimilarity between latent spaces learned by different models. Finally, to specifically quantify the preservation of local temporal structure, we implemented a Temporal Neighborhood Preservation analysis. For each data point, we identified its k=10 nearest neighbors in the temporal domain (by frame index) and its k=10 nearest neighbors in the t-SNE latent space (by Euclidean distance). The similarity between these two sets of neighbors was measured using the Jaccard index, and the score was averaged across all points in the sequence.

$$Loss_{\text{focal}} = \alpha(1 - p)^{\gamma}|y - \hat{y}| \tag{4}$$

$$Loss_{\text{FFT}} = \left\| |\mathcal{F}(y)| - |\mathcal{F}(\hat{y})| \right\|_1 \tag{5}$$

$$Loss_{\text{SSIM}} = 1 - \text{SSIM}(y, \hat{y}) \tag{6}$$

$$Loss_{\text{perceptual}} = \|\phi(y) - \phi(\hat{y})\|_2^2 \tag{7}$$

## K  DECODER ARCHITECTURE SELECTION

To validate the architectural choice of our visual decoder, we conducted a comparative analysis between our proposed U-Net decoder and a standard Transformer-based decoder. To ensure a fair

comparison, the Transformer decoder was capacity-matched (i.e., approximately equal number of total parameters) to our U-Net implementation.

The results revealed a significant performance gap: the Transformer decoder achieved a Movie SSIM of $\approx 0.48$, substantially lower than the 0.593 achieved by our U-Net decoder. This performance difference highlights the importance of the spatial inductive bias inherent in convolutional architectures (U-Net) for dense pixel prediction tasks. While Transformers excel at modeling long-range dependencies, they lack the intrinsic local connectivity required for high-fidelity image reconstruction from sparse neural embeddings, particularly in the limited-data regime of biological recordings. Consequently, we adopted the U-Net architecture as the optimal choice for our decoding framework.

## L    Task-Specific Neural Representation Analysis

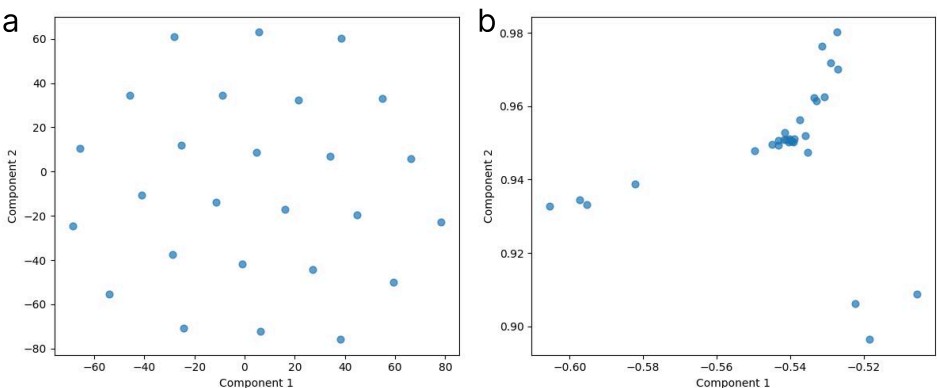

Figure S5: **Task-specific adaptation of the learned latent manifold.** Visualization of the final latent representations from our fine-tuned model on two different tasks. **(a) Drifting Gratings:** For this classification task, the model learns a geometrically structured representation with distinct, well-separated clusters corresponding to the 8 stimulus directions. **(b) Movie Decoding:** For this reconstruction task, the model learns a continuous, non-linear manifold that captures the temporal trajectory of the movie frames.

The learned representations for the two main downstream tasks exhibit fundamentally different geometries, as confirmed by a high Procrustes disparity (0.95) and low Centered Kernel Alignment (CKA, 0.18). This demonstrates that our pretrained model does not use a rigid, one-size-fits-all representation, but rather adapts its internal structure to the specific demands of each task. For the *drifting gratings* classification task (Figure S5a), the model learns to organize its representations into discrete, maximally separated clusters to optimize for classification. In contrast, for the *movie decoding* reconstruction task (Figure S5b), it learns a continuous, non-linear manifold that effectively represents the temporal flow of the visual experience. This adaptability highlights the model's ability to learn the true underlying structure of a given neural decoding problem.

## M    Pretrained Knowledge is Distributed Along Encoder Components

To investigate where pretrained information is stored, we conducted ablation experiments by selectively freezing encoder components during finetuning. Our results reveal that the learned representation is distributed, not localized. Partially freezing any single component led to catastrophic reconstruction failures, whereas surprisingly, freezing the *entire* encoder better preserved spatial content (Figure S6). This suggests that the pretrained representation relies on coordinated interactions across the entire encoder and requires holistic, rather than modular, adaptation during fine-tuning.

Figure S6: Encoder freezing analysis. Freezing any subset of encoder layers degrades high-frequency detail, indicating pretrained knowledge is distributed across the entire encoder. **(a)** Ground truth, **(b)** ours (did not freeze encoder), **(c)** full encoder freezing, **(d)** partial encoder freezing (former layers only). **(e)** partial encoder freezing (middle layers only). **(f)** partial encoder freezing (latter layers only).

# N   USE OF LARGE LANGUAGE MODELS IN MANUSCRIPT PREPARATION

We acknowledge the use of a large language model (Google's Gemini) for language editing and refinement during the preparation of this manuscript. The model was employed to improve grammar, clarity, and conciseness. The authors meticulously reviewed and revised all model-generated suggestions to ensure scientific accuracy and preserve the original meaning. All conceptual work, experimental results, and scientific conclusions presented herein are entirely the work of the authors.

