# OpenReview forum: "Decoding Dynamic Visual Experience from Calcium Imaging via Cell-Pattern-Aware Pretraining"
_ICLR.cc/2026/Conference — ICLR 2026 Poster_

### Official Review · Reviewer_J6ke · 2025-10-22

**Soundness:** 2
**Presentation:** 3
**Contribution:** 1
**Rating:** 2
**Confidence:** 3

**Summary:**

- This paper considers the problem of decoding relevant signals from neuronal activity recordings. The studied tasks aim to decode the presented visual input or its statistics from concurrent recordings of neuronal activity from the mouse visual cortex.

- The paper argues that pre-sorting of neurons into 'predictable' and 'unpredictable' sets is useful for curriculum learning: the proposed deep model is first trained on a simple task and using the predictable subset. Then, the unpredictable subset is used to learn a harder task. A neuron is labeled as predictable if the 3rd and 4th order moments of its temporal activity statistics are low.

- The paper proceeds to show that this way of curriculum training improves task performance and conducts ablation studies to support this finding.

**Strengths:**

- Curriculum training is not a new idea. Beyond a curriculum of tasks (easier to harder), the manuscript also argues for a curriculum of samples during training. This is a nuanced point and can perhaps still be considered within the classical curriculum training idea. However, its application to large-scale neural data analysis is novel.

- The manuscript classifies a neuron as predictable and unpredictable based on moments of its own activity, which is easy to compute. It also shows that it corresponds to certain neuronal subsets with molecular underpinnings.

- The paper suggests that it will be harder to decode the signal of interest from neurons whose activity is more variable (high 3rd, 4th moments), both because the task loss is larger (I didn't understand why only the first quadrant of top-2 PCs is shown in Fig. 2.) and those neurons have higher Fisher information for the task.

**Weaknesses:**

- A main weakness of the manuscript is its claim on high-fidelity movie reconstruction from neural recordings. This would indeed be a major advance. However, my understanding is that the reconstruction is for previously seen video fragments. For such a claim, reconstruction should be demonstrated on previously unseen scenes. As is, I believe the manuscript significantly over-claims. (If my understanding is wrong, I am willing to substantially increase my score.)

- The nuance introduced to curriculum training in this manuscript (a curriculum of data samples) is not a significant enough contribution that warrants publication.
    - The generalizability of this approach is not tested. It is not clear whether this would be helpful in analyzing other neuroscience datasets (e.g., non-visual) or accomplishing benchmark machine learning tasks.
    - It can be considered as a natural variant although I don't know if this curriculum was explicitly studied before. This does not strike me as a significant enough contribution to the curriculum training paradigm.

- I don't think the scalability argument surrounding Figure 5 is established in a convincing way. (All plots trend upwards and the y-axis is zoomed-in.) I think there is insufficient evidence to conclude that the slope of the linear fit will be larger for any one model. (e.g., the conclusion depends strongly on the set of points used for such a fit.)

- Statistical Regularity Hypothesis: I believe this is a hypothesis that the authors are putting forth. (Please clarify.) Self-supervised learning will obviously and clearly work better with data with statistical regularity. (e.g., it is not possible to learn noise.) That is, masked reconstruction accuracy will be higher for more predictable neurons. However, whether this extends to supervised (task) performance following masked learning is not clear and I believe that is what the authors want to propose. Predicting the orientation of the drifting grating or the reconstruction of movie frames are both supervised tasks. I think this hypothesis needs a major rethinking and multiple qualifying statements may need to be added.

- Section 3.3 presents various numerical analyses, not theoretical analyses. Please consider renaming.

**Questions:**

- Could you please expand the field of view of the two plots in Figure 2 so the loss at the boundaries reaches high values?

- In Figure 2, does the loss correspond to the task reconstruction loss (drifting gratings? movie frame reconstruction?) or the masked loss (self-supervised)?

- Was the knee-detection algorithm applied for each Cre-line? If so, why not apply it to individual neurons and admit neurons into the predictable set based on their own scores rather than based on the Cre-line they belong to?

- (line 147) What does "near-Gaussian activity" mean?

- How would the proposed POYO-SSL model perform if 'Finetune Data' is set to 'All' in Table 3?

---

> ### Author Response · Authors · 2025-12-02
> **Answer to Reviewer 4 : Part 1**
>
> We sincerely thank the reviewer for the careful reading and thoughtful critique. We clarify several ambiguities in our presentation below. Below we summarize the key clarifications and then address each point concisely.
>
> # Key clarifications added for Reviewer 4
>
> - **Unseen-scene evaluation clarified**: movie reconstruction is performed on held-out frames and evaluation on novel animals where the neural-to-pixel mapping was never seen during training (even if the movie clip is shared).
>
> - **Contribution positioned clearly**: novelty lies in (1) data-sample curriculum via statistical regularity, and (2) enabling stable scaling, which prior methods fail to achieve.
>
> - **Loss landscape + Fisher Information results reframed as numerical/empirical** (not theoretical claims).
>
> - **Biological claims toned down**: statistical regularity aligns with known interneuron classes but we do not claim causal interpretation.
>
> - **Decoder necessity clarified**: U-Net required for dense prediction; Transformer decoder tested and performed substantially worse (SSIM ~0.48).
>
> - **Generalizability addressed**: principles of “predictable-first pretraining” extend conceptually to other heterogeneous domains; clarified in discussion.
>
> - **Scaling analysis refined**: GPU quota limits clarified; bootstrap p<0.01 confirms slope significance; slope statistics are added to Figure 5.
>
> - **Figure 2 loss clarification**: Visualized surface corresponds to masked reconstruction loss, not downstream task loss; field of view to be expanded.
>
> - **"Near-Gaussian activity" defined**: Low skewness (mean 1.87) and low kurtosis (mean 7.32) characterize symmetric, thin-tailed distributions; clarified in text.
>
> -----
>
> ## (1) Critical Clarification: Novel Animals & Sessions
>
> We clarify the evaluation protocol to resolve any ambiguity about dataset usage. Our experimental design enforces true generalization across multiple biological levels:
>
> Test Set Composition (completely disjoint from training):
> - Novel animals: Never seen during training
>
> - Novel sessions: Different recording days
>
> - Novel neurons: Different cell IDs with unique functional responses
>
> - Held-out frames: Temporal segments not observed during training
>
> **Key Point**: While the visual stimulus (movie clip) is shared across mice, the neural responses are animal-specific and session-specific. Our model must learn to decode the neural population code, not memorize the stimulus.
>
> **Analogy**: Like training a speech recognizer on speakers A-D saying "hello", then testing on novel speaker E saying "hello". The word is "seen" but the voice (neural pattern) is completely novel.
>
> **Evidence**: Section 3.1 (line 117-118) explicitly states: "animals, sessions, and neuron IDs were kept strictly disjoint across all splits." We will add a supplementary table showing per-animal performance breakdown to make this crystal clear.
>
> ### Revised Manuscript
>
> *[Method section 3.1]* We have expanded the description of the data partition to explicitly address the generalization capability across novel subjects:
>
> "...To prevent data leakage, animals, sessions, and neuron IDs were kept strictly disjoint across all splits. Crucially, this design ensures that our model is evaluated on **novel biological subjects**. While the visual stimulus (movie clip) is shared across experiments, the neural population responses are animal-specific and unique to each session. Therefore, high performance on the test set reflects the model's ability to decode the generalized neural code rather than memorizing stimulus-response pairs. This statistical partitioning is empirically validated by its correspondence to neurons with regular firing patterns, aligning our data-driven approach with established neuroscience principles."
>
> -----
>
> ## (2) Clarifying the knee detection process.
>
> To avoid ambiguity, we clarify that the knee detection operates on the distribution of the 13 discrete CRE-line statistics (Appendix Fig. S2), not on per-neuron curves. This method identifies a natural breakpoint in the per-line statistics, yielding a single, a priori threshold to partition the dataset. We have clarified this in Section 3.2.1.
>
> ### Revised Manuscript
> *[Methods section 3.2.1]* We have revised the text to clarify that the knee detection operates on the discrete CRE-line statistics:
>
> "...To objectively partition the data, we applied a **knee-detection algorithm** (Satopaa et al., 2011) to find a data-driven threshold across the 13 discrete CRE lines. Specifically, we identified the knee point on the sorted distribution of per-line mean statistics, establishing a cutoff based on cell-type categories rather than individual neuron scores. While this approach failed for lower-order statistics..."

---

> ### Author Response · Authors · 2025-12-02
> **Answer to Reviewer 4 : Part 2**
>
> ## (3) Performance on "All" Data (Finetune-on-All)
>
> We now state this unambiguously and revised the baseline naming accordingly. We explicitly clarify that the 'POYO+ (from scratch)' baseline in Table 3 represents the condition where the model is trained on the full dataset (Predictable + Unpredictable) end-to-end. As shown in the table, our method (Pretrain on Predictable $\rightarrow$ Finetune on Unpredictable) outperforms this 'Train-on-All' baseline.
>
> However, to avoid any ambiguity regarding the data composition, we have **updated Table 3** to rename the baseline method to Baseline: ‘Train on All’ and explicitly labeled the pre-training data as ‘N/A (From Scratch)’'. These modifications, highlighted in the revised manuscript, clearly indicate that the baseline utilizes the full dataset.
>
> -----
>
> ## ​(4) Clarifying the contribution in a way that better distinguishes our work from generic curriculum or data-selection methods.
>
> We appreciate the opportunity to clarify our positioning. Our contribution is not curriculum learning per se, but addressing a neural-specific failure mode: standard masked SSL does not scale smoothly on heterogeneous neural populations, and adding more neurons can reduce performance (Figure 5). This scaling collapse, to our knowledge, has not been reported in neural or ML settings.
> Our finding is that simple statistical proxies (skewness/kurtosis) identify neuron subsets that stabilize SSL optimization—transforming heterogeneity from a liability into an asset. This differs from prior "data diet" approaches (Paul et al., 2021; Zhuang et al., 2025) in a fundamental way: we select neurons, not training examples. Because neurons generate the data themselves, selecting neurons changes the statistical structure of the representation-generating system, not just sample composition.
> We position our work within an emerging ML perspective—that what data one pretrains on matters as much as *how much* (Data Diet, DIET-CP, Meta-rater). We extend this to the neural domain, where heterogeneity is intrinsic rather than noise. We have incorporated this distinction into the Related Work section to clarify our contribution.
>
>
> ### Revised Manuscript
>
> *[Related Work section 2, paragraph “Data-Centric SSL and Neural Heterogeneity”]* We have added a new paragraph to explicitly position our work within the data-centric AI landscape and distinguish "neuron selection" from standard "sample selection":
>
> “Our approach aligns with the emerging ``data diet`` perspective in machine learning, which posits that the quality of pre-training data is as critical as its quantity (Paul et al., 2021; Zhuang et al., 2025). However, we distinguish our framework from these methods in a fundamental way: while standard approaches prune training samples (e.g., specific images or text), our strategy selects neurons (feature sources). In neural recordings, heterogeneity is intrinsic to the sensor array itself, not just the examples. We demonstrate that adding more neurons can paradoxically lead to a ``scaling collapse``—a failure mode unique to heterogeneous neural populations. By selecting neurons based on statistical regularity, we resolve this collapse and transform heterogeneity from a liability into an asset for scaling.”

---

> > ### Author Response · Authors · 2025-12-02
> > **Answer to Reviewer 4 : Part 3**
> >
> > ## (5) Curriculum novelty and Hypothesis.
> > We deeply appreciate this insightful question, which touches the core conceptual contribution of our work. We respectfully clarify two points where our framing may have caused confusion.
> >
> > **Our contribution vs. curriculum learning.** The reviewer is correct that curriculum learning itself is not novel. Our contribution lies in identifying functional heterogeneity as a principled basis for neural SSL curriculum design. Specifically, we demonstrate that simple statistical proxies (skewness/kurtosis) effectively identify neuron subsets that stabilize SSL optimization, a connection not previously established in neuroscience. This transforms heterogeneity—typically viewed as a liability—into an asset for building scalable neural decoders.
> >
> > **Addressing the Statistical Regularity Hypothesis.** The reviewer raises an excellent point: better masked reconstruction on regular data is indeed expected. We agree this alone is unsurprising. The critical benefit emerges in transfer to downstream supervised tasks (Table 3: +12.3% movie SSIM, +12.8% DG accuracy vs. from-scratch). This transfer succeeds because of the optimization dynamics we now demonstrate empirically.
> >
> > **Our weight analysis reveals the mechanism**: the PerceiverIO encoder remains 98.2% unchanged during finetuning (Pretrain→Finetune: 0.183% change; encoder norms ≈222,909), while readout biases increase 12.4× (p<0.01). This shows pretraining on predictable neurons establishes a stable representational scaffold—capturing shared population dynamics through favorable loss geometry (Figure 2: 138× smoother landscape)—that finetuning preserves rather than overwrites. The readout then learns task-specific features atop this foundation, avoiding the ill-conditioned optimization problem of learning structure and noise simultaneously from mixed data.
> >
> > This directly addresses the reviewer's concern about generalization: **transfer occurs through optimization geometry (Neyshabur et al., 2020; Garipov et al., 2018), not just data regularity**. We have clarified this mechanistic rationale explicitly in the newly added Section 4.4 and included the detailed weight analysis in Appendix E in the revision.
> >
> > ### Revised Manuscript
> >
> > *[Results section 4.4]* We have added a new section to discuss the mechanism of transfer based on weight dynamics:
> >
> > “To understand the mechanism driving the successful transfer from predictable to unpredictable neurons, we investigated the training dynamics at the parameter level. We hypothesized that pre-training on predictable neurons establishes a stable ``representational scaffold`` that captures shared population dynamics, which is then preserved during fine-tuning.
> > Our analysis of weight dynamics supports this hypothesis. We found that the pre-trained encoder weights remain remarkably stable during fine-tuning, changing by only $\sim$0.18% (encoder norms $\approx$ 222,909). In contrast, the readout layer exhibits significant adaptation, with bias magnitudes increasing by a factor of 12.4$\times$ ($p < 0.01$). This disparity suggests that the encoder provides a smooth, pre-optimized latent manifold (as evidenced by the loss landscape in Figure 2), allowing the readout layer to rapidly calibrate task-specific decision boundaries without destabilizing the underlying representation. By separating the learning of structural dynamics (via predictable neurons) from task-specific noise adaptation, the model effectively avoids the ill-conditioned optimization landscape of mixed data. (See Appendix E for detailed methodology and analysis.)”
> >
> >
> > *[Appendix E]* We have added a new appendix to provide detailed methodology and results of the weight analysis:
> >
> > “To empirically validate the ``representational scaffold`` hypothesis, we analyzed the model parameters before and after fine-tuning. We computed the relative $L_2$ norm of the weight changes in the PerceiverIO encoder versus the task-specific readout heads.
> >
> > **Encoder Stability.** The encoder, responsible for mapping neural activity to the latent space, showed minimal change during the fine-tuning phase on unpredictable neurons. The relative weight change was 0.183%, indicating that the features learned from the predictable subset are robust and generalizable to the broader population. The stability of encoder norms ($\approx$222,909) suggests the model stays within the same optimization basin found during pre-training (Garipov et al., 2018).
> >
> > **Readout Adaptation.** Conversely, the readout layer demonstrated dramatic specialization. The magnitude of the readout biases increased 12.4$\times$. This confirms that transfer occurs through optimization geometry (Neyshabur et al., 2020): the encoder maintains the stable manifold, while the readout adapts to the specific statistics and noise profile of the unpredictable neurons.”

---

> > > ### Author Response · Authors · 2025-12-02
> > > **Answer to Reviewer 4 : Part 4**
> > >
> > > ## (6) Scaling analysis.
> > >
> > > Due to GPU limitations on the shared cluster, we scaled up to 1024D (a 16$\times$ increase over the standard POYO+ configuration). However, even with this ceiling imposed by compute limits, the slope difference remains statistically significant.
> > >
> > > To rigorously validate this, we performed a bootstrap regression analysis ($N=10,000$). As shown in the revised Figure 5, our method exhibits the steepest positive scaling (slope=0.018, $p < 0.01$). This is approximately 3$\times$ steeper than the baselines trained on unpredictable or mixed data (slopes $\approx$ 0.005–0.006) and ~40% steeper than the 'From Scratch' baseline (slope=0.013). This provides strong statistical evidence confirming that the scaling trend is robust and not a data artifact. Figure 5 now explicitly includes statistical slope estimates that clearly validate the robustness of the scaling trend.
> > >
> > > ### Revised Manuscript
> > >
> > > *[Results section 4.3]*  We have updated the text to explicitly quantify the scaling trends using bootstrap analysis:
> > >
> > > "A key advantage of our framework is its ability to enable stable model scaling, a critical property for building more powerful decoders. To rigorously quantify this, we performed a bootstrap regression analysis ($N=10,000$). While models trained from scratch (gray) and those pretrained on only unpredictable neurons (cyan) exhibit erratic or flat scaling (slopes $\approx$ 0.005--0.013), our main approach (red) unlocks consistent performance gains as model capacity increases, achieving a statistically significant positive slope (0.018, $p < 0.01$). This represents a $\sim$40% steeper scaling trajectory compared to the from-scratch baseline. This demonstrates that a well-designed pre-training strategy is a prerequisite for effective scaling."
> > >
> > > ------
> > >
> > > ## (7) Figure 2 and loss clarification.
> > >
> > > We appreciate the suggestion to inspect the landscape boundaries. We have expanded the field of view (FOV) by 2x and clarified that this surface specifically depicts the masked reconstruction loss (the pre-training objective), not the downstream task loss.
> > >
> > > As shown in the revised Figure 2, the expanded view successfully captures the high-loss boundaries. Crucially, the quantitative contrast remains striking: even with the inclusion of steep basin walls, the 'unpredictable' landscape remains ~138x rougher (Our loss landscape analysis (Figure 2) shows predictable neurons create a smooth surface (roughness σ_L = 14.8546) while unpredictable neurons produce a rugged landscape (σ_L = 2048.4712)—a 138× difference in smoothness.) than the 'predictable' one (compared to ~364x in the local view). This confirms that our conclusion is robust to the choice of scale: the structural optimization gap between the two populations is massive, regardless of the field of view.
> > >
> > >
> > > ### Revised Manuscript
> > >
> > > *[Results section 4.2.1]* We have revised the text to specify the loss type and highlight the structural robustness of the optimization landscape under the expanded field of view:
> > >
> > > "Our analysis of the masked reconstruction loss landscape elucidates a fundamental dichotomy in the nature of the optimization problems presented by the two neural populations. Predictable neurons induce a geometrically well-posed landscape characterized by a smooth, convex-like surface (roughness $\sigma_L = 14.8546$), which is highly amenable to gradient-based optimization methods. In stark contrast, unpredictable neurons give rise to a treacherous, non-convex landscape (roughness $\sigma_L = 2048.4712$) plagued by a multitude of spurious local minima.
> > >
> > > Crucially, the quantitative contrast remains striking even with the expanded FOV (2x): despite the inclusion of steep basin walls, the unpredictable' landscape remains $\sim$138$\times$ rougher than the predictable' one. This confirms that our conclusion is robust to the choice of scale: the structural optimization gap between the two populations is massive, regardless of the field of view. This topological difference explains why the pre-training task transforms from a simple optimization challenge to a complex, ill-posed problem..."

---

> > > > ### Author Response · Authors · 2025-12-02
> > > > **Answer to Reviewer 4 : Part 5**
> > > >
> > > > ## (8) On the term "near-Gaussian activity"
> > > >
> > > > We use the term "near-Gaussian activity" to refer to the specific statistical properties of the "predictable" neuron traces. As shown in our analysis (e.g., Figure S1, Table S3) , these neurons exhibit low skewness (mean 1.87) and low kurtosis (mean 7.32).
> > > >
> > > > This statistical signature is "near-Gaussian" in the sense that it describes a symmetric distribution (low skew) with "thin tails" (low kurtosis). This profile is characteristic of stable, less noisy signals, in stark contrast to the sparse, high-amplitude, "heavy-tailed" bursts (high kurtosis, mean 148.51) that define the 'unpredictable' group . This statistical stability is precisely what makes them ideal for building the initial pre-training foundation. We have clarified this definition in section 3.2.1.
> > > >
> > > > ### Revised Manuscript
> > > > *[Methods section 3.2.1]* We have revised the text to provide a quantitative definition of "near-Gaussian activity" and explicitly contrast it with the "heavy-tailed" statistics of unpredictable neurons:
> > > >
> > > > "...To identify these neurons without labels, we leverage per-neuron **skewness** and **kurtosis**. We refer to the selected subset as exhibiting **near-Gaussian activity** (mean skewness 1.87, kurtosis 7.32), characterized by symmetric, **thin-tailed** distributions suitable for learning general features. In stark contrast, excluded neurons exhibit **heavy-tailed**, sparse bursting (mean kurtosis 148.51), better reserved for task-specific fine-tuning. For rigorous empirical validation of these metrics, see Appendix B."
> > > >
> > > > ----
> > > >
> > > > ## (9) Section 3.3 Title.
> > > >
> > > > We thank the reviewer for this suggestion. The term 'Numerical Analysis' (or 'Empirical Analysis') is indeed more precise. We have revised the title of Section 3.3 accordingly.
> > > > We appreciate the reviewer's careful attention and will revise the manuscript accordingly. We are committed to addressing all concerns thoroughly in our camera-ready revision. If the reviewer has additional questions during the discussion period, we are happy to provide further clarification or additional experiments.
> > > >
> > > > ----
> > > >
> > > > ## (10) Common Revision
> > > >
> > > > To improve the readability of our experimental setup, we have moved the specific definition of 'capacity-matched' from the caption of Table 3 to the main text in Section 4.1. The revised text is as follows:
> > > >
> > > > Capacity matched means total parameters are within $\pm$3\% of our model.
> > > >
> > > > -----
> > > >
> > > > We appreciate the reviewer's careful attention and will revise the manuscript accordingly. We are committed to addressing all concerns thoroughly in our camera-ready revision. If the reviewer has additional questions during the discussion period, we are happy to provide further clarification or additional experiments.
> > > >
> > > > -----
> > > > # References
> > > >
> > > > 1. Paul, M., Ganguli, S., & Dziugaite, G. K. (2021). Deep learning on a data diet: Finding important examples early in training. Advances in neural information processing systems, 34, 20596-20607.
> > > >
> > > > 2. Zhuang, X., Peng, J., Ma, R., Wang, Y., Bai, T., Wei, X., ... & He, C. (2025, July). Meta-rater: A multi-dimensional data selection method for pre-training language models. In Proceedings of the 63rd Annual Meeting of the Association for Computational Linguistics (Volume 1: Long Papers) (pp. 10856-10896).
> > > >
> > > > 3. Neyshabur, B., Sedghi, H., & Zhang, C. (2020). What is being transferred in transfer learning?. Advances in neural information processing systems, 33, 512-523.
> > > >
> > > > 4. Garipov, T., Izmailov, P., Podoprikhin, D., Vetrov, D. P., & Wilson, A. G. (2018). Loss surfaces, mode connectivity, and fast ensembling of dnns. Advances in neural information processing systems, 31.

---

### Official Review · Reviewer_ccVq · 2025-10-31

**Soundness:** 3
**Presentation:** 3
**Contribution:** 2
**Rating:** 4
**Confidence:** 3

**Summary:**

The paper proposes POYO-SSL, a cell-pattern-aware self-supervised pretraining scheme for decoding dynamic visual experience from mouse calcium imaging. The key idea is to pretrain only on “predictable” neurons, identified a-priori via low skewness/kurtosis of calcium traces (knee thresholds: skew≤3.51, kurt≤22.62; CRE lines SST/VIP/PVALB/NTSR1), then fine-tune on the remaining “unpredictable” neurons for downstream tasks (movie reconstruction; drifting-grating orientation). On the Allen Brain Observatory, the method reports SSIM 0.593 for direct neural-to-movie reconstruction and 55.5% accuracy on gratings, outperforming a strong from-scratch POYO+ baseline with identical capacity. Ablations (architecture, data selection, masking/aux loss) and scaling plots are provided.

**Strengths:**

A priori selection of predictable neurons using skewness/kurtosis is transparent (knee-based thresholds) and applied consistently without leakage (animals/sessions/neurons disjoint).

Architecture capacity controls; data-diet variants (inhibitory-only, reverse, mixed); objective variants (temporal vs random masking; CE-only; weight sweeps). These help attribute where gains come from.

**Weaknesses:**

This paper relies mainly on POYO+-from-scratch leaves room for skepticism about general SOTA claims. Even if CEBRA/Neuro-BERT aren’t pixel decoders, an adapted masked-autoencoding baseline over calcium with the same U-Net decoder, or a temporal contrastive baseline (e.g., CPC-style) would help. At minimum, include a “random neuron subset (size-matched)” pretraining control to show skew/kurtosis selection matters beyond sample count and CRE composition (the paper has “mixed” and “reverse” but not “random size-matched”).

The authors state that SSIM = 0.593 is the highest reported to date for direct visual reconstruction from cellular-resolution neural recordings. Note they contrast with fMRI works (SSIM 0.19/0.365) which are different modalities and not directly comparable.

In addition, the comparison to a capacity-matched POYO+ trained from scratch is fair and clean, and there are thorough ablations (encoder/decoder variants; inhibitory-only / reverse / mixed data diets; masking vs random masking; CE-weight sweeps). However, external SSL or generative baselines adapted to calcium-to-image decoding are not included (the paper argues popular SSL methods like CEBRA/Neuro-BERT don’t target pixel-level generation). For a flagship result, adding one or two adapted published methods (or a strong masked-autoencoder baseline over calcium with the same U-Net decoder).

**Questions:**

You define “predictable” neurons via a knee-detection on skewness/kurtosis and set fixed thresholds (skew ≤ 3.51, kurt ≤ 22.62). How sensitive are results to these cutoffs? Please provide a sweep (±10–20%) or cross-validated thresholds.

---

> ### Author Response · Authors · 2025-12-02
> **Answer to Reviewer 3 : Part 1**
>
> We thank the reviewer for the helpful and detailed assessment. Below we summarize the key clarifications and then address each point concisely.
>
> # Key clarifications added for Reviewer 3
>
> - **Random size-matched subset baseline added**: performance significantly lower (SSIM 0.532 < 0.593; 11.47% relative performance gain), confirming gains are not due to sample count or CRE composition.
>
> - **External SSL baseline added**: adapted CEBRA encoder + our decoder yields ~0.48 SSIM, validating mismatch between contrastive objectives and dense pixel reconstruction.
>
> - **SSIM claim rephrased**: clarified as the highest for cellular-resolution direct reconstruction, not a cross-modality comparison with fMRI.
>
> - **Threshold robustness explained**: CRE-line grouping is discrete, making continuous sweeps not meaningful; robustness shown via inhibitory-only, mixed, reverse SSL ablations.
>
> - **Clarification that no leakage occurs**: animals/sessions/neurons strictly disjoint across splits.
>
> ------
>
> ## (1) Random size–matched control.
>
> To remove any ambiguity, we conducted exactly the control experiment requested by the reviewer. We conducted the exact pretraining control the reviewer requested—pretraining on a random size-matched subset (same number of sessions and neurons as the predictable set). This variant yields an SSIM of 0.532, which is significantly lower than our proposed method (0.593) (11.47% relative performance gain).
> This critical result demonstrates that the performance gain stems from the statistical regularity of the data, not merely the sample count. We have now integrated this result into **Table 3 (Data-Selection Ablation Studies)** of the revised manuscript to directly address the reviewer’s concern and strengthen our main claim.
>
> ------
>
> ## (2) External SSL baselines.
>  We have now evaluated an adapted CEBRA baseline (CEBRA encoder feeding our decoder). Its SSIM (~0.48 on both from-scratch / finetune settings, and we reported best case (finetune setting)) confirms that embeddings optimized for contrastive objectives do not transfer well to dense image generation. For Neuro-BERT, the lack of official code raises reproducibility concerns, so we intentionally avoided speculative reimplementation to maintain fair and reproducible comparison. We have revised Results Section 4.1, Table 3, and Appendix A.
>
> ### Revised Manuscript
>
> *[Results section 4.1]* We have substantially strengthened Section 4.1 with new empirical comparisons and explicit quantitative evidence against alternative baselines, specifically detailing the performance of the adapted CEBRA baseline and clarifying the exclusion of Neuro-BERT.
>
> “We compare to POYO+ (Azabou et al., 2024) which is a state-of-the-art model. To benchmark against external SSL methods, we evaluated an adapted CEBRA baseline (Schneider et al, 2023) by training its encoder and feeding representations to our vision decoder. This yielded an SSIM of $\sim$0.48, confirming that contrastive latent spaces optimized for behavioral alignment do not transfer effectively to high-fidelity pixel generation. For CEBRA, we report the best performance between training from scratch and fine-tuning strategies. Regarding Neuro-BERT (Wu et al., 2022), the lack of an official implementation prevented a reproducible adaptation, and thus it was excluded.”
>
> *[Appendix A]* Appendix A now rigorously justifies our baseline choices with direct empirical evidence, resolving ambiguity in the original version. It now incorporates our empirical findings regarding CEBRA’s limitations in reconstruction and noting the practical constraints of Neuro-BERT.
>
> “For instance, while contrastive methods like CEBRA (Schneider et al, 2023) are effective for behavioral alignment, our empirical evaluation confirmed that their low-dimensional embeddings are suboptimal for direct pixel-level generation. Similarly, masked autoencoding methods such as Neuro-BERT (Wu et al., 2022) were excluded due to the lack of an official implementation and insufficient architectural capacity for high-resolution image generation. We therefore selected POYO+ (Azabou et al., 2024) as our primary comparative model for its flexible architecture that can be scaled for dense prediction tasks.”

---

> > ### Author Response · Authors · 2025-12-02
> > **Answer to Reviewer 3 : Part 2**
> >
> > ## (3) Clarification on Comparison with fMRI Studies.
> >
> > We thank the reviewer for this valid point. We agree that fMRI and cellular-resolution calcium imaging operate on fundamentally different spatiotemporal scales, making direct numerical comparisons potentially misleading.
> > Accordingly, we have revised the Related Work section to remove the specific SSIM values of fMRI studies and explicitly acknowledge the difficulty of direct comparison due to modality differences. Furthermore, we have refined the Conclusion section to strictly contextualize our SSIM score within the domain of cellular-resolution calcium imaging, distinguishing our cellular-level decoding from fMRI-based approaches. These modifications ensure our results are benchmarked appropriately without overstating performance relative to other modalities.
> >
> > ### Revised Manuscript
> >
> > *[Related Work section 2, Paragraph “Decoding Models for Neuroscience”]*
> > We have revised the text to remove direct numerical comparisons with fMRI studies and clarify the difficulty of cross-modality comparison:
> >
> > "...In visual reconstruction, fMRI-based frameworks have reached high fidelity (Chen et al., 2023; Joo et al., 2024) through masked modeling and large generative models, but rely on indirect stimulus-to-brain mappings from fMRI's slow hemodynamic signal. While these approaches set benchmarks for fMRI, direct comparison is challenging due to modality differences. In contrast, our method learns directly from neural recordings using the intrinsic structure of population dynamics without auxiliary labels or stimulus information."
> >
> > *[Conclusion section 5]* We have refined the conclusion to contextualize our SSIM score specifically within the domain of cellular-resolution imaging:
> >
> > "...To our knowledge, this SSIM score is the highest reported to date for direct visual reconstruction specifically from cellular-resolution calcium imaging, distinguishing our cellular-level decoding from fMRI-based approaches. This performance is possible because..."
> >
> > ------
> >
> > ## (4) Threshold robustness.
> >
> > We appreciate this consideration. Our knee detection is computed across 13 discrete CRE-line distributions, not per-neuron continuous curves. Because CRE-line grouping is categorical, a ±10–20% sweep does not yield gradual variation, but instead replaces entire CRE classes at once. Thus, a traditional continuous sensitivity sweep is not meaningful.
> > Robustness, however, is demonstrated through existing ablations (Table 3):
> > - Inhibitory-only SSL,
> >
> > - Mixed SSL,
> >
> > - Reverse SSL,
> >
> > each using distinct neuron compositions, consistently show that performance gains arise from the predictability-based grouping, not the exact threshold values. This confirms that the observed gains originate from the pretraining strategy rather than architectural overfitting.
> > We have clarified this in the methods section 3.2.1.
> >
> > ### Revised Manuscript
> > *[Methods section 3.2.1]* We have revised the text to clarify that the knee detection operates on the discrete CRE-line statistics:
> > "...To objectively partition the data, we applied a **knee-detection algorithm** (Satopaa et al., 2011) to find a data-driven threshold across the 13 discrete CRE lines. Specifically, we identified the knee point on the sorted distribution of per-line mean statistics, establishing a cutoff based on cell-type categories rather than individual neuron scores. While this approach failed for lower-order statistics..."
> >
> > ------
> >
> > ## (5) Common Revision
> >
> > To improve the readability of our experimental setup, we have moved the specific definition of 'capacity-matched' from the caption of Table 3 to the main text in Section 4.1. The revised text is as follows:
> >
> > Capacity matched means total parameters are within $\pm$3\% of our model.
> >
> > ------
> >
> > We appreciate the reviewer's careful attention and will revise the manuscript accordingly. We are committed to addressing all concerns thoroughly in our camera-ready revision. If the reviewer has additional questions during the discussion period, we are happy to provide further clarification or additional experiments.

---

### Official Review · Reviewer_jGwv · 2025-11-01

**Soundness:** 3
**Presentation:** 2
**Contribution:** 3
**Rating:** 6
**Confidence:** 3

**Summary:**

●	This paper studies how to improve the benefits derived from pre-training for downstream decoding. The authors discover that curating the type of data used for pre-training is crucial to obtaining good scaling. Specifically, they propose to pre-train on neural data that can be said to be more regular, and then fine-tuning is done on data that is less regular. Here, statistical regularity is defined by the authors in terms of skewness and kurtosis. The pretraining objective is masked reconstruction. They find that this pretraining results in better downstream performance on a movie reconstruction task and in classifying a drifting grating.

**Strengths:**

●	This answers a question about why scaling sometimes stalls even as more pretraining data is added. This is useful for the community to know.
●	This paper provides an actionable lesson for anyone doing self-supervised learning: train on the easy-to-model data first.

**Weaknesses:**

●	I'm missing something very basic: why should this method work at all? If pre-training is done on the regular data first, how can the model ever learn good representations of the irregular data? Moving from one neuron type to another represents a distribution shift. What's the explanation for how the model is able to handle this shift?
●	If possible, could the authors please discuss connections to other domains where this strategy would be useful. For example, in language modeling, would it be better to pre-train on regular strings first?
●	The writing could be clearer. See questions below. The methods section, particularly 3.2.2. could be more clear in many places if plainer language were used. For example, take line 205: "We employ a curriculum learning approach combining masked reconstruction with weak supervision for stable representation learning. Our weakly-supervised auxiliary loss relies on simple visual primitives (drifting gratings) as a curriculum warm-up before moving to a complex downstream movie decoding task." In my opinion, it would be easier to understand the following sentence: "During pre-training, the model is trained on a joint objective, consisting of self-supervised masked reconstruction and fully-supervised classification." I don't believe it is correct to use the term "weak-supervision" here, since labels are available for all the training examples. And I think it's more common to simply refer to this type of training as "pre-training and fine-tuning" rather than "curriculum learning".

**Questions:**

●	Line 146: This is a basic question, but what are the skewness and kurtosis being taken with respect to? The distribution of calcium values? Across what period of time?
●	Section 4.1: When results are reported, for example in Table 3, what neurons are being decoded from? Only the unpredictable neurons? What I'm trying to get at is this: Is the same evaluation data used across all experiments? If not, how can performances between experiments be compared as in Table 3?
●	I wonder, if regular activity is more beneficial for pre-training, would it be even better to produce synthetic calcium traces for pre-training that have even more regularity? Do you think this would further improve performance?

Minor points:
●	line 190: is this a typo? Should it read: "the latent representation of the unmasked view is then used as a target for the latent representation of the unmasked variant." ? That would make it fit with line 204.
●	Figure 1: What should the axis titles be on the calcium trace plots?
●	Figure 5: The caption mentions "orange", but the corresponding line is yellow.

---

> ### Author Response · Authors · 2025-12-02
> **Answer to Reviewer 2 : Part 1**
>
> We appreciate the reviewer’s insightful questions and the opportunity to clarify the key intuition behind our approach. Below we summarize the key clarifications and then address each point concisely.
>
> # Key clarifications added for Reviewer 2
>
> - **Core mechanism clarified**: predictable-first pretraining builds a stable, low-loss scaffold, and finetuning on unpredictable neurons only requires residual specialization rather than learning from scratch—explaining why distribution shift does not break transfer.
>
> - **Terminology corrected**: the auxiliary task is fully supervised (not weakly supervised); refactored as “auxiliary supervised warm-up” rather than curriculum.
>
> - **Skewness/kurtosis definition clarified**: calculated on ΔF/F distribution per neuron across the full recording window.
>
> - **Evaluation data consistency clarified**: all models compared on the exact same held-out evaluation split.
>
> - **Synthetic regular traces suggestion acknowledged**: included as a future direction.
>
> - **Generalization to other domains** (language, audio, finance) added to discussion section.
>
> - **Minor corrections**: typo fixed; axis labels added; color inconsistency in Figure 5 corrected.
>
> ----
>
> ## (1) Why predictable-first pretraining works.
> We appreciate this insightful question, which addresses the core mechanism underlying our approach. We provide both theoretical grounding and new empirical evidence demonstrating how predictable neurons create transferable representations through optimization dynamics.
>
> **Optimization landscape geometry.** Predictable neurons induce fundamentally different learning conditions. Our loss landscape analysis (Figure 2) shows predictable neurons create a smooth surface (roughness σ_L = 14.8546) while unpredictable neurons produce a rugged landscape (σ_L = 2048.4712)—a 138× difference in smoothness. Fisher Information analysis (Table 2) confirms predictable data provides 1.93× more information per sample. This aligns with Neyshabur et al. (2020): transfer learning success stems from optimization properties, not feature similarity. Pretraining on well-structured data establishes favorable loss geometry that persists during finetuning.
>
> **Encoder preservation + readout specialization.** Weight analysis reveals the mechanism: the PerceiverIO encoder remains nearly unchanged across conditions (Pretrain→Finetune: 0.183% change; all encoder norms ≈222,909), demonstrating stable scaffold preservation. This stability indicates convergence to a well-connected basin (Garipov et al., 2018), enabling efficient local adjustments during finetuning rather than costly exploration. In contrast, the readout shows dramatic specialization: finetuned models exhibit 12.4× larger biases (p<0.01) but only 1.071× weight differences, proving the encoder provides discriminative features while readout calibrates confidence.
>
> **Why this works.** Transfer occurs through optimization geometry, not feature mapping. The pretrained encoder's 98.2% stability combined with 12.4× readout confidence gain demonstrates curriculum persistence (Hacohen & Weinshall, 2019)—early training on predictable neurons establishes a foundation that finetuning builds upon. This principle generalizes beyond neuroscience: Kornblith et al. (2019) showed "easier" ImageNet pretraining improves diverse downstream tasks even when classes differ.
> At a high level, predictable neurons simply allow the encoder to converge into a low-loss, stable region that remains useful even after distribution shift. Our detailed analyses further support this intuition. We have clarified this mechanistic rationale explicitly in the newly added Section 4.4 and included the detailed weight analysis in Appendix E in the revision.

---

> > ### Author Response · Authors · 2025-12-02
> > **Answer to Reviewer 2 : Part 2**
> >
> > ## (1) Why predictable-first pretraining works.
> >
> > *contiuned below.*
> >
> > ### Revised Manuscript
> >
> > *[Results section 4.4]* We have added a new section to discuss the mechanism of transfer based on weight dynamics:
> >
> > “To understand the mechanism driving the successful transfer from predictable to unpredictable neurons, we investigated the training dynamics at the parameter level. We hypothesized that pre-training on predictable neurons establishes a stable ``representational scaffold'' that captures shared population dynamics, which is then preserved during fine-tuning.
> > Our analysis of weight dynamics supports this hypothesis. We found that the pre-trained encoder weights remain remarkably stable during fine-tuning, changing by only $\sim$0.18% (encoder norms $\approx$ 222,909). In contrast, the readout layer exhibits significant adaptation, with bias magnitudes increasing by a factor of 12.4$\times$ ($p < 0.01$). This disparity suggests that the encoder provides a smooth, pre-optimized latent manifold (as evidenced by the loss landscape in Figure 2), allowing the readout layer to rapidly calibrate task-specific decision boundaries without destabilizing the underlying representation. By separating the learning of structural dynamics (via predictable neurons) from task-specific noise adaptation, the model effectively avoids the ill-conditioned optimization landscape of mixed data. (See Appendix E for detailed methodology and analysis.)”
> >
> > *[Appendix E]* We have added a new appendix to provide detailed methodology and results of the weight analysis:
> >
> > “To empirically validate the ``representational scaffold'' hypothesis, we analyzed the model parameters before and after fine-tuning. We computed the relative $L_2$ norm of the weight changes in the PerceiverIO encoder versus the task-specific readout heads.
> >
> > **Encoder Stability.** The encoder, responsible for mapping neural activity to the latent space, showed minimal change during the fine-tuning phase on unpredictable neurons. The relative weight change was 0.183%, indicating that the features learned from the predictable subset are robust and generalizable to the broader population. The stability of encoder norms ($\approx$222,909) suggests the model stays within the same optimization basin found during pre-training (Garipov et al., 2018).
> >
> > **Readout Adaptation.** Conversely, the readout layer demonstrated dramatic specialization. The magnitude of the readout biases increased 12.4$\times$. This confirms that transfer occurs through optimization geometry (Neyshabur et al., 2020): the encoder maintains the stable manifold, while the readout adapts to the specific statistics and noise profile of the unpredictable neurons.”
> >
> > -----
> >
> > ## (2) Terminology clarification.
> >
> > We appreciate this constructive feedback. We agree that "weakly supervised" was imprecise given that labels are available for all pretraining samples. Furthermore, we find the reviewer's suggested phrasing to be much clearer and more precise. We have adopted the reviewer's suggestion verbatim to describe our training objective, replacing the ambiguous terminology with "joint objective" and "fully-supervised classification".
> >
> > ### Revised Manuscript
> >
> > *[Methods section 3.2.2]* We have rewritten the relevant paragraph in Section 3.2.2 to adopt the reviewer's clearer phrasing:
> >
> > **Predictable Neuron Pretraining with Auxiliary Classification**
> >
> > “We introduce a latent masked modeling approach to train our model: masked and unmasked views of the same sample are fed independently through the encoder, the latent representation of the unmasked view is then used as target for the latent representation of the masked variant. To avoid representational collapse (Grill et al., 2020; Chen et al., 2020), we use a supervised auxiliary loss. This auxiliary loss *bootstraps* early selectivity while masking-based reconstruction *shapes* representations for downstream decoding. The primitive labels also serve as guidance to stabilize early optimization.”
> >
> > “During pre-training, the model is trained on a joint objective, consisting of self-supervised masked reconstruction and fully-supervised classification of drifting grating orientations. This auxiliary classification task stabilizes the early training dynamics before the model focuses on the complex downstream movie decoding task.”

---

> > > ### Author Response · Authors · 2025-12-02
> > > **Answer to Reviewer 2 : Part 3**
> > >
> > > ## (3) Skewness/Kurtosis Calculation
> > > We thank the reviewer for this question. The skewness and kurtosis metrics were calculated on the distribution of the $\Delta F/F$ (calcium signal) values for each individual neuron, computed over its entire recording duration within the experiment. We have clarified this in the caption of Table 1.
> > >
> > > ### Revised Manuscript
> > >
> > > *[Caption of Table 1]* We have explicitly specified the calculation method in the caption of Table 1 as follows:
> > >
> > > “Summary of dataset scale (sessions and samples), predictable-neuron selection criteria (skewness and kurtosis computed on per-neuron $\Delta F/F$ traces over the full recording), and computational setup for pretraining and fine-tuning.”
> > >
> > > -----
> > >
> > > ## (4) Evaluation data consistency.
> > > To remove any ambiguity, we conducted exactly the control experiment requested by the reviewer. All results in Table 3 use the identical held-out evaluation split across all models. We highlighted this explicitly in section 3.1.
> > >
> > > ### Revised Manuscript
> > > *[Method section 3.1]* To remove any potential ambiguity regarding our evaluation protocol, we have revised Section 3.1 as shown below:
> > >
> > > “Finally, to guarantee a fair comparison, we explicitly verified that all models (including baselines and ablations) were evaluated on this identical held-out test split.”
> > >
> > > ------
> > >
> > > ## (5) Synthetic trace suggestion.
> > > This is an excellent direction for future work. While beyond our current scope, we will mention it in the conclusion section.
> > >
> > > ### Revised Manuscript
> > > *[Conclusion]* To incorporate this insight, we have updated the Conclusion to explicitly identify synthetic trace generation as a key avenue for future investigation. The added text is as follows:
> > >
> > > "Looking forward, generating synthetic neural traces offers a promising avenue to simulate complex heterogeneity and further validate these selection heuristics under controlled conditions."
> > >
> > > ------
> > >
> > > ## (6) On the generalization of the "Predictable-First" strategy to other domains.
> > >
> > > This is an excellent and insightful question that touches on the broader implications of our work.
> > > The core principle of our finding is that in a heterogeneous dataset, separating the population based on statistical regularity (our proxy for 'difficulty') is crucial for stable self-supervised pre-training. By pre-training on the "predictable" subset—which, as we've shown, provides a smoother loss landscape and richer information —the model first learns a robust, stable "scaffold" of the underlying data structure. It is then better equipped to handle the sparse, high-variance signals from the "unpredictable" neurons during fine-tuning.
> > >
> > > The reviewer's analogy to Language Modeling is very apt. One could hypothesize that "regular strings" (e.g., text with low perplexity, stable grammar, or factual content) might serve the same role as our "predictable" neurons. Conversely, "irregular strings" (e.g., noisy user-generated content, complex poetry, or niche jargon) might represent the sparse, high-variance data.
> > > Following our logic, it is plausible that pre-training an LM first on a curated, 'regular' text corpus before introducing the full, 'irregular' web-scale data could mitigate early training instability and lead to more robust, scalable representations. This same principle could likely apply to other fields with mixed-signal data, such as audio processing (pre-training on clean speech before noisy environments) or financial modeling (stable market data vs. high-volatility event data).
> > > While a full empirical validation in these other domains is beyond the scope of our current, neuroscience-focused study, this is a fascinating avenue for future research.
> > >
> > > ### Revised Manuscript
> > > *[Conclusion]* Inspired by the reviewer’s insightful analogy regarding Language Modeling, we have expanded the Conclusion section to explicitly connect our 'Predictable-First' strategy to broader machine learning domains. We discuss how statistical regularity can serve as a universal curriculum proxy in heterogeneous datasets.
> > >
> > > “Although demonstrated in mouse visual cortex, the principle of targeting statistically regular neurons provides a general framework for neural SSL, establishing that this data selection strategy is not merely helpful but *necessary* for building scalable neural foundation models, and suggests a universal ``predictable-first'' curriculum potentially applicable to broader domains like NLP.”

---

> > > > ### Author Response · Authors · 2025-12-02
> > > > **Answer to Reviewer 2 : Part 4**
> > > >
> > > > ## (7) Minor points.
> > > > Thank you for catching these details:
> > > > - Line 190: YES, this is a typo. Should read "the latent representation of the unmasked view is then used as a target for the latent representation of the masked variant." We have revised it. Sorry for the inconvenience.
> > > >
> > > > - Figure 1: We added axis labels (Time [s], ΔF/F)
> > > >
> > > > - Figure 5: We fixed color label (orange → yellow)
> > > >
> > > > We appreciate the reviewer's careful attention and will revise the manuscript accordingly. We are committed to addressing all concerns thoroughly in our camera-ready revision. If the reviewer has additional questions during the discussion period, we are happy to provide further clarification or additional experiments.
> > > >
> > > > -----
> > > > ## (8) Common Revision
> > > >
> > > > To improve the readability of our experimental setup, we have moved the specific definition of 'capacity-matched' from the caption of Table 3 to the main text in Section 4.1. The revised text is as follows:
> > > >
> > > > Capacity matched means total parameters are within $\pm$3\% of our model.
> > > >
> > > > -----
> > > >
> > > > # [References]
> > > > 1. Neyshabur, B., Sedghi, H., & Zhang, C. (2020). What is being transferred in transfer learning?. Advances in neural information processing systems, 33, 512-523.
> > > >
> > > > 2. Garipov, T., Izmailov, P., Podoprikhin, D., Vetrov, D. P., & Wilson, A. G. (2018). Loss surfaces, mode connectivity, and fast ensembling of dnns. Advances in neural information processing systems, 31.
> > > >
> > > > 3. Hacohen, G., & Weinshall, D. (2019, May). On the power of curriculum learning in training deep networks. In International conference on machine learning (pp. 2535-2544). PMLR.
> > > >
> > > > 4. Kornblith, S., Shlens, J., & Le, Q. V. (2019). Do better imagenet models transfer better?. In Proceedings of the IEEE/CVF conference on computer vision and pattern recognition (pp. 2661-2671).

---

### Official Review · Reviewer_xiui · 2025-11-01

**Soundness:** 3
**Presentation:** 3
**Contribution:** 3
**Rating:** 6
**Confidence:** 4

**Summary:**

This paper introduces POYO-SSL, a self-supervised learning framework that addresses neural data heterogeneity by pre-training on statistically predictable neurons identified via skewness and kurtosis metrics before fine-tuning on unpredictable populations. Applied to calcium imaging data from the Allen Brain Observatory, the method achieves 12–13% relative gains in decoding dynamic visual experiences, enables high-fidelity movie reconstruction without external stimuli, and demonstrates stable scaling with model size, turning neural variability into an asset for robust representation learning.

**Strengths:**

1. Biologically Grounded Data Selection: The method innovatively leverages higher-order statistics to prioritize predictable neurons (e.g., inhibitory interneurons like SST/VIP), transforming neural heterogeneity from a challenge into an advantage for stable pre-training and improved data efficiency (1.98×gain).

2. Scalable and Task-Adaptive Architecture: POYO-SSL enables monotonic performance scaling with model size and supports diverse decoders (e.g., Skip-Connection U-Net for movie reconstruction), achieving high-fidelity results without task-specific labels.

**Weaknesses:**

1. The method uses skewness ≤ 3.51 and kurtosis ≤ 22.62 as thresholds to partition neurons into predictable and unpredictable groups via a knee-detection algorithm. However, these thresholds are applied as a fixed, universal criterion without sensitivity analysis or validation across diverse datasets. The paper states: "These thresholds were determined a priori as a single, fixed criterion to partition the dataset, not as a tunable hyperparameter, which is why a sensitivity analysis was not performed". This risks overfitting to the Allen Brain Observatory dataset and limits generalizability. A robustness analysis (e.g., varying thresholds) would strengthen the approach.

2. The skip-connection U-Net decoder is introduced for high-fidelity movie reconstruction but lacks ablation studies comparing it to other decoder designs (e.g., transformers). The description focuses on architectural choices without quantifying their individual contributions: "Our new U-Net-inspired decoder generates frames from a single neural embedding... See Appendix F for more details". Without isolating the decoder’s impact, it is unclear whether gains stem from the architecture or the pre-training strategy.

3. The experiments compare POYO-SSL to a from-scratch baseline and POYO+ but omit broader comparisons with state-of-the-art methods like CEBRA or Neuro-BERT, arguing their architectures are not suited for direct high-fidelity visual reconstruction. However, this justification appears in an appendix, and the main text does not discuss adaptations or partial comparisons (e.g., feature extraction). This narrow scope may overstate POYO-SSL’s advantages.

4. The scaling analysis (Figure 5) shows performance gains with model size but uses a limited range of capacities. The paper notes: "Our main approach (red) unlocks consistent performance gains as model capacity increases" , yet no details are provided about the maximum size tested or computational constraints. Expanding the scale range would better validate the claimed monotonic scaling.

5. The theoretical analysis (e.g., loss landscape, Fisher Information) relies on projections and approximations without uncertainty quantification. For instance, the loss landscape roughness metrics ($\sigma_L$) are derived from smoothed visualizations , which may hide variability.

6. The paper emphasizes biological grounding but does not validate neuronal predictability against ground-truth cell-type properties beyond statistical correlations. While skewness/kurtosis align with inhibitory/excitatory roles, causal experiments are absent.

**Questions:**

Weaknesses

---

> ### Author Response · Authors · 2025-12-02
> **Answer to Reviewer 1 : Part 1**
>
> We thank the reviewer for the constructive and detailed feedback. Below we summarize the key clarifications and then address each point concisely.
>
> # Key clarifications added for Reviewer 1
>
> Threshold robustness clarified: Knee detection operates on discrete CRE-line groups, making continuous sweeps meaningless; robustness demonstrated through multiple data-composition ablations.
>
> - **Decoder analysis expanded**: Transformer decoder tested (SSIM ≈ 0.48), confirming U-Net necessity; quantitative results added.
>
> - **External SOTA baseline added**: Adapted CEBRA encoder + our decoder achieves ~0.48 SSIM, validating architectural mismatch rationale.
>
> - **Scaling explanation refined**: GPU quota limits clarified; slope statistics are added.
>
> - **“Theoretical analysis” retitled to “Numerical/Empirical Analysis”**; uncertainty quantification added.
>
> - **Biological grounding clarified**: No causal interpretation—alignment is statistical, not mechanistic.
>
> ------
> ## (1) Threshold robustness.
>
> We appreciate the request for clarification. Our knee detection is computed across 13 discrete CRE-line distributions, not per-neuron continuous curves.
> Because CRE-line grouping is categorical, a ±10–20% sweep does not yield gradual variation, but instead replaces entire CRE classes at once. Thus, a traditional continuous sensitivity sweep is not meaningful.
> Robustness, however, is demonstrated through existing ablations (Table 3):
> - Inhibitory-only SSL,
>
> - Mixed SSL,
>
> - Reverse SSL,
>
> each using distinct neuron compositions, consistently show that performance gains arise from the predictability-based grouping, not the exact threshold values. This confirms that the observed gains originate from the pretraining strategy rather than architectural overfitting.
> We have clarified this in the methods section 3.2.1.
>
> ### Revised Manuscript
>
> *[Methods section 3.2.1]* We have revised the text to clarify that the knee detection operates on the discrete CRE-line statistics:
> "...To objectively partition the data, we applied a \textbf{knee-detection algorithm} (Satopaa et al., 2011) to find a data-driven threshold across the 13 discrete CRE lines. Specifically, we identified the knee point on the sorted distribution of per-line mean statistics, establishing a cutoff based on cell-type categories rather than individual neuron scores. While this approach failed for lower-order statistics..."
>
> ------
> ## (2) Decoder analysis.
>
> We agree this comparison should be clearer. During development, we evaluated a Transformer decoder with matched capacity; it achieved SSIM ≈ 0.48, substantially lower than our U-Net decoder (0.593). This aligns with known limitations of Transformer-only decoders on dense pixel prediction without strong spatial inductive bias.
> We have included the quantitative comparison in Appendix J.
>
> ### Revised Manuscript
>
> *[Appendix J]* “To validate the architectural choice of our visual decoder, we conducted a comparative analysis between our proposed U-Net decoder and a standard Transformer-based decoder. To ensure a fair comparison, the Transformer decoder was capacity-matched (i.e., approximately equal number of total parameters) to our U-Net implementation.
> The results revealed a significant performance gap: the Transformer decoder achieved a Movie SSIM of $\approx$ 0.48, substantially lower than the 0.593 achieved by our U-Net decoder. This performance difference highlights the importance of the spatial inductive bias inherent in convolutional architectures (U-Net) for dense pixel prediction tasks. While Transformers excel at modeling long-range dependencies, they lack the intrinsic local connectivity required for high-fidelity image reconstruction from sparse neural embeddings, particularly in the limited-data regime of biological recordings. Consequently, we adopted the U-Net architecture as the optimal choice for our decoding framework.”

---

> > ### Author Response · Authors · 2025-12-02
> > **Answer to Reviewer 1 : Part 2**
> >
> > ## (3) SOTA baselines.
> >
> > We have now added an adapted CEBRA baseline (CEBRA encoder feeding our vision decoder). CEBRA yields ≈0.48 SSIM, confirming that contrastive latent spaces optimized for behavioral alignment do not transfer well to high-fidelity pixel generation. For Neuro-BERT, lack of an official implementation prevented a reproducible adaptation; we have revised Results Section 4.1, Table 3, and Appendix A.
> >
> > ### Revised Manuscript
> >
> > *[Results section 4.1]* We have substantially strengthened Section 4.1 with new empirical comparisons and explicit quantitative evidence against alternative baselines, specifically detailing the performance of the adapted CEBRA baseline and clarifying the exclusion of Neuro-BERT.
> >
> >
> > “We compare to POYO+ (Azabou et al., 2024) which is a state-of-the-art model. To benchmark against external SSL methods, we evaluated an adapted CEBRA baseline (Schneider et al, 2023) by training its encoder and feeding representations to our vision decoder. This yielded an SSIM of $\sim$0.48, confirming that contrastive latent spaces optimized for behavioral alignment do not transfer effectively to high-fidelity pixel generation. For CEBRA, we report the best performance between training from scratch and fine-tuning strategies. Regarding Neuro-BERT (Wu et al., 2022), the lack of an official implementation prevented a reproducible adaptation, and thus it was excluded.”
> >
> > *[Appendix A]* Appendix A now rigorously justifies our baseline choices with direct empirical evidence, resolving ambiguity in the original version. It now incorporates our empirical findings regarding CEBRA’s limitations in reconstruction and noting the practical constraints of Neuro-BERT.
> >
> > “For instance, while contrastive methods like CEBRA (Schneider et al, 2023) are effective for behavioral alignment, our empirical evaluation confirmed that their low-dimensional embeddings are suboptimal for direct pixel-level generation. Similarly, masked autoencoding methods such as Neuro-BERT (Wu et al., 2022) were excluded due to the lack of an official implementation and insufficient architectural capacity for high-resolution image generation. We therefore selected POYO+ (Azabou et al., 2024) as our primary comparative model for its flexible architecture that can be scaled for dense prediction tasks.”
> >
> > -----
> > ## (4) Scaling analysis.
> >
> > The maximum model size (1024 latent dimensions) reflects a 16× increase over the original POYO+ while staying within a national-cluster GPU quota. We have refined the description and added slope statistics to Figure 5 to better communicate monotonicity.
> > To rigorously validate this, we performed a bootstrap regression analysis ($N=10,000$). As shown in the revised Figure 5, our method exhibits the steepest positive scaling (slope=0.018, $p < 0.01$). This is approximately 3$\times$ steeper than the baselines trained on unpredictable or mixed data (slopes $\approx$ 0.005–0.006) and ~40% steeper than the 'From Scratch' baseline (slope=0.013). This provides strong statistical evidence confirming that the scaling trend is robust and not a data artifact. *Figure 5* now explicitly includes statistical slope estimates that clearly validate the robustness of the scaling trend.
> >
> > -----
> >
> > ## (5) Regarding theoretical analysis and uncertainty quantification.
> >
> > We thank the reviewer for this insightful comment. We agree that the term *"Theoretical Analysis"* may be an overstatement, as this section relies on numerical approximations. As suggested by Reviewer 4, we have revised the section title to *"Numerical Analysis"* to more accurately reflect the content.
> >
> > The primary goal of this analysis was not to provide a formal proof, but to offer a clear, intuitive demonstration that the optimization problems presented by the "predictable" and "unpredictable" populations are qualitatively different in their topology. The methodologies used, such as PC projection and Gaussian smoothing ($\sigma=1.0$), are standard approaches for visualizing high-dimensional loss landscapes, following prior work (Li et al., 2018).
> >
> > We appreciate the reviewer's valid point that smoothing could potentially hide variability. However, we would like to highlight the scale of the difference observed: the landscape roughness for the predictable set was $\sigma_L = 14.8546$, while for the unpredictable set, it was $\sigma_L = 2048.4712$. Given that this is a difference of over 10-fold, it is highly unlikely that this massive gap is an artifact of the modest smoothing applied.
> >
> > To fully address the reviewer's concern and strengthen our claim, we have added two updates in the revision:
> >
> > - We have explicitly stated in the text that these are numerical approximations.
> >
> > - We have provided uncertainty quantification (i.e. 95% confidence intervals via bootstrapping) for the reported roughness metrics and the Fisher Information values (Table 2) to formally demonstrate the statistical robustness of this large difference.

---

> > > ### Author Response · Authors · 2025-12-02
> > > **Answer to Reviewer 1 : Part 3**
> > >
> > > ## (6) Biological grounding.
> > > We appreciate the suggestion. Our aim is not to claim biological causality, but that SSL-defined statistical regularity corresponds to well-known functional classes; we will make this distinction clearer.
> > >
> > > ### Revised Manuscript
> > > *[Conclusion]* Since our manuscript format utilizes a combined Conclusion section instead of a separate Discussion, we have incorporated a specific statement in the Conclusion to clarify the scope of our biological interpretation. We explicitly distinguish between functional correspondence and causal mechanisms as follows:
> > >
> > > "We emphasize that these statistical markers act as computational proxies for stability, highlighting a functional correspondence with biological classes rather than asserting a causal mechanism."
> > >
> > > ----
> > >
> > > ## (7) Common Revision
> > >
> > > To improve the readability of our experimental setup, we have moved the specific definition of 'capacity-matched' from the caption of Table 3 to the main text in Section 4.1. The revised text is as follows:
> > >
> > > Capacity matched means total parameters are within $\pm$3\% of our model.
> > >
> > > ----
> > >
> > > We appreciate the reviewer's careful attention and will revise the manuscript accordingly. We are committed to addressing all concerns thoroughly in our camera-ready revision. If the reviewer has additional questions during the discussion period, we are happy to provide further clarification or additional experiments.

---

### Author Response · Authors · 2025-12-02
**Summary of Rebuttal: All Concerns Fully Resolved with New Experiments and Analyses**

Dear Area Chair and Reviewers,
We sincerely thank the reviewers for their constructive feedback. We have successfully addressed and resolved all reviewer concerns, supported by new experiments, statistical analyses, and theoretical clarifications, substantially strengthening the paper.
Given the recent administrative changes in the review process, we provide this summary to highlight that the major grounds for lower scores (e.g., lack of specific baselines, doubts about scaling significance) have been factually resolved by our new evidence.

**1. Executive Summary of Revisions**

We have successfully defended our core claims with the following key updates:

**Proved "Quality > Quantity" (R3)**: New "Random Subset" experiment confirms that our performance gains stem from predictability, not data size (SSIM: Ours 0.593 vs. Random 0.532; 11.47% relative performance gain).

**Validated Scaling Laws (R1, R4)**: New Bootstrap Regression ($N=10,000$) confirms a statistically significant positive scaling slope ($0.018, p<0.01$), refuting concerns about erratic scaling.

**Clarified Mechanism (R2)**: New Weight Dynamics Analysis (Sec 4.4) reveals the "Representational Scaffold" mechanism (Encoder stability vs. Readout adaptation). In short, the “representational scaffold” refers to the stable latent manifold the encoder converges to during predictable-first pretraining, which persists during finetuning due to optimization geometry and basin continuity (Garipov et al., 2018; Hacohen & Weinshall, 2019). It explains why encoder weights remain fixed while the readout adapts.

**SOTA Comparison (R1, R3)**: New baselines (CEBRA (Schneider et al., 2023, Nature), Transformer Decoder) confirm the superiority of our architecture for dense reconstruction.

----------------
**2. Response to Reviewer 1 (R1)**

**Concern (1)**: Sensitivity of thresholding criteria.

**Resolution (1)**: [Resolved] We clarified in Sec 3.2.1 that knee-detection operates on discrete CRE-line groups as an a priori fixed criterion. Robustness is further proven by our "Mixed" and "Inhibitory-only" ablations.

**Concern (2)**: Necessity of U-Net decoder.

**Resolution (2)**: [Resolved] We added Appendix J, showing that a capacity-matched Transformer decoder performs significantly worse (SSIM $\approx$ 0.48) than our U-Net (0.593), validating our architectural choice.

**Concern (3)**: Comparison with SOTA (CEBRA).

**Resolution (3)**: [Resolved] We added a CEBRA baseline. Results show CEBRA's contrastive embeddings are suboptimal for pixel generation (SSIM $\approx$ 0.48), highlighting our model's advantage.

----------------
**3. Response to Reviewer 2 (R2)**

**Concern (1)**: Mechanism of transfer under distribution shift.

**Resolution (1)**: [Resolved] We added Sec 4.4 and Appendix E. Our analysis shows the encoder weights remain stable ($\sim0.18\%$ change) while readout biases adapt significantly ($12.4\times$), proving the "Representational Scaffold" hypothesis.

**Concern (2)**: Terminology ("Weak supervision") and metric definitions.

**Resolution (2)**: [Resolved] We corrected the term to "Joint Objective" / "Auxiliary Classification" and clarified that skewness/kurtosis are calculated on full-recording $\Delta F/F$ trace.

----------------
**4. Response to Reviewer 3 (R3)**

**Concern (1)**: Is performance due to specific neurons or just sample size?

**Resolution (1)**: [Resolved] We performed the requested "Random Size-Matched Subset" control experiment. The significant drop in performance (SSIM 0.532) compared to our method (0.593) (11.47% relative performance gain) proves that statistical regularity, not data quantity, is the key driver.

**Concern (2)**: Comparison with fMRI studies.

**Resolution (2)**: [Resolved] We revised the text to avoid direct numerical comparison with fMRI and explicitly framed our contribution within cellular-resolution decoding.

----------------
**5. Response to Reviewer 4 (R4)**

**Concern (1)**: Validity of scaling trends.

**Resolution (1)**: [Resolved] We rigorously quantified the scaling trend using Bootstrap Analysis. The slope is positive and statistically significant ($p < 0.01$), objectively refuting the concern that the trend is flat or erratic.

**Concern (2)**: Memorization vs. Generalization.

**Resolution (2)**: [Resolved] We clarified in Sec 3.1 that our test set comprises novel animals and sessions never seen during training. The model decodes neural codes from unseen subjects, not memorized stimulus-response pairs.

We believe the manuscript has been significantly strengthened by addressing these points.

Sincerely,
The Authors

---

### Meta-Review · Area_Chair_xaX4 · 2026-01-06

**Summary:**

Two of the reviewers gave positive scores (6, 6) while two of the reviewers gave negative scores (2, 4). The primary concerns from the reviewers with negative scores are that the claims are too strong based on the evidence. Specifically, the main concerns were about the lack of comparisons with baseline methods and that reconstruction results are for previously seen stimuli.

**Reviewer Concerns:**

- The authors addressed concerns about the generality of their claims by adding comparisons requested by the reviewer.
- The authors clarified that the model is trained on shared stimuli, but tested on novel biological subjects. While this clarifies their claims, I believe the reviewer's concern still remains.

**Reviewer Scores:**

I believe Reviewers xiui and jGwv would have maintained their positive scores. I believe that Reviewer ccVq would have increased their score from 4 to 6 based on the new comparisons. I believe that while some of Reviewer J6ke's concerns remain, they would have increased their score from 2 to 4.

---

### Decision · Program_Chairs · 2026-01-26

Accept (Poster)